# Anionic phospholipids control mechanisms of GPCR-G protein recognition

Naveen Thakur [1,8], Arka P. Ray[1,8], Liam Sharp[2], Beining Jin[1], Alexander Duong[1], Niloofar Gopal Pour[1], Samuel Obeng[3,4], Anuradha V. Wijesekara[1], Zhan-Guo Gao[5], Christopher R. McCurdy [4,6], Kenneth A. Jacobson [5], Edward Lyman[2,7] & Matthew T. Eddy [1] ✉

G protein-coupled receptors (GPCRs) are embedded in phospholipids that strongly influence drug-stimulated signaling. Anionic lipids are particularly important for GPCR signaling complex formation, but a mechanism for this role is not understood. Using NMR spectroscopy, we explore the impact of anionic lipids on the function-related conformational equilibria of the human $A_{2A}$ adenosine receptor ($A_{2A}AR$) in bilayers containing defined mixtures of zwitterionic and anionic phospholipids. Anionic lipids prime the receptor to form complexes with G proteins through a conformational selection process. Without anionic lipids, signaling complex formation proceeds through a less favorable induced fit mechanism. In computational models, anionic lipids mimic interactions between a G protein and positively charged residues in $A_{2A}AR$ at the receptor intracellular surface, stabilizing a pre-activated receptor conformation. Replacing these residues strikingly alters the receptor response to anionic lipids in experiments. High sequence conservation of the same residues among all GPCRs supports a general role for lipid-receptor charge complementarity in signaling.

The essential roles of lipids as key regulators of the structures and functions of membrane proteins are well established[1–7]. For the 826 human G protein-coupled receptors (GPCRs), mounting evidence documented in the literature has increasingly highlighted the critical roles of lipids in regulating ligand-induced cellular signaling, both through their bulk physical properties, such as membrane curvature[8], and as specific chemical partners acting as orthosteric ligands[9,10] or as allosteric modulators[11–13].

Recent structural and biophysical studies point to special roles of anionic lipids in regulating GPCR activity. Biochemical data indicated anionic lipids strongly influenced selectivity of the β₂-adrenergic receptor (β₂AR) for different G proteins through lipid-protein charge complementarity[14], and anionic lipids impacted the affinity and efficacy of β₂AR ligands[11]. Anionic lipids, including POPS, were found to be critical for activation of the CB₂ cannabinoid receptor[15]. Mass spectrometry data supported the role of the anionic phospholipid PtdIns(4,5)P2 (PIP2) stabilizing complexes of human GPCRs with their partner G proteins[16]. Anionic phospholipids such as PIP2 have also been observed bound to the human serotonin receptor 5-HT₁ₐ in cryo-electron microscopy structures[17], supporting the idea that direct lipid-protein interactions modulate receptor activity.

Knowledge of GPCR structural plasticity is critical to developing an understanding of signaling mechanisms. Nuclear magnetic resonance (NMR) spectroscopy is well-suited to experimentally investigate

[1]Department of Chemistry, College of Liberal Arts & Sciences, University of Florida, 126 Sisler Hall, Gainesville, FL, USA. [2]Department of Physics and Astronomy, University of Delaware, Newark, Delaware, USA. [3]Department of Pharmacodynamics, College of Pharmacy, University of Florida, Gainesville, FL, USA. [4]Department of Medicinal Chemistry, College of Pharmacy, University of Florida, Gainesville, FL, USA. [5]Laboratory of Bioorganic Chemistry, National Institute of Diabetes and Digestive and Kidney Diseases, National Institutes of Health, Bethesda, Maryland 20892, USA. [6]Translational Drug Development Core, Clinical and Translational Sciences Institute, University of Florida, Gainesville, Florida, USA. [7]Department of Chemistry and Biochemistry, University of Delaware, Newark, Delaware, USA. [8]These authors contributed equally: Naveen Thakur, Arka P. Ray. ✉e-mail: matthew.eddy@chem.ufl.edu

and measure GPCR structural plasticity, as it provides the unique capability to detect multiple, simultaneously populated receptor conformations and link observed conformational equilibria to efficacies of bound drugs[18,19]. We leveraged this capability to investigate the impact of anionic lipids on the structural plasticity of the human A$_{2A}$ adenosine receptor (A$_{2A}$AR), a representative class A GPCR and exemplary GPCR for investigating receptor-lipid interactions. Our investigation builds on accumulated data from NMR spectroscopic studies of A$_{2A}$AR interactions with small molecule ligands[12,20–26], providing a firm foundation for evaluating the impact of anionic lipids. In both mass spectrometry experiments[16] and molecular dynamics simulations[27], anionic lipids were observed to strongly influence the formation of A$_{2A}$AR signaling complexes. However, a mechanistic basis for these observations has not been determined.

Using nanodiscs, we precisely controlled phospholipid composition to investigate the response of A$_{2A}$AR over a wide range of defined binary lipid mixtures. In $^{19}$F-NMR data, we observed the impact of anionic lipids on the conformational equilibria of A$_{2A}$AR to be of a similar magnitude as bound drugs. Synergy observed between the presence of anionic lipids and efficacy of bound drugs indicated that sensitivity to lipid composition depended on the receptor conformation. In both NMR experiments and computational models, positively charged residues on the A$_{2A}$AR intracellular surface near regions that interact with partner signaling proteins appeared to facilitate the influence of anionic lipids. Integrating these data with correlative signaling assays and NMR experiments of judiciously selected A$_{2A}$AR variants provided a mechanistic view on the role of lipids in A$_{2A}$AR signaling. Key residues identified in these experiments are conserved among not only class A but all receptor classes, indicating this mechanism may be shared across the GPCR superfamily.

## Results

### Molecular recognition of drug compounds by A$_{2A}$AR investigated across a wide range of phospholipid compositions

We expressed human A$_{2A}$AR containing a single extrinsic cysteine introduced into position 289, A$_{2A}$AR[A289C] (Supplementary Fig. 1). The extrinsic cysteine, located near the intracellular-facing surface of transmembrane (TM) helix VII, was introduced for $^{19}$F-NMR experiments, and this A$_{2A}$AR variant was previously shown to retain pharmacological activity of the native receptor[22,28]. Purified A$_{2A}$AR[A289C] was reconstituted into lipid nanodiscs formed with the membrane scaffold protein MSP1D1[29] containing defined binary mixtures of different molar ratios of zwitterionic phospholipids, POPC (1-palmitoyl-2-oleoyl-glycero-3-phosphocholine) or POPE (1-palmitoyl-2-oleoyl-sn-glycero-3-phosphoethanolamine), and anionic lipids, POPS (1-palmitoyl-2-oleoyl-sn-glycero-3-phospho-L-serine), POPA (1-palmitoyl-2-oleoyl-sn-glycero-3-phosphate), POPG (1-palmitoyl-2-oleoyl-sn-glycero-3-phospho-(1'-rac-glycerol)) or PI(4,5)P$_2$ (1,2-dioleoyl-sn-glycero-3-phospho-(1'-myo-inositol-4',5'-biphosphate)). Analytical characterization of purified lipid nanodiscs containing A$_{2A}$AR[A289C] showed highly monodispersed and homogenous samples (Supplementary Fig. 1). $^{31}$P-NMR in aqueous solutions was used to verify the lipid composition within the nanodisc samples studied in subsequent biophysical and NMR spectroscopy experiments (Supplementary Fig. 2). As the lipid headgroups showed resolved signals with unique chemical shifts, the intensities of the observed $^{31}$P signals were used to quantify the relative amounts of each lipid species present. For all samples and for all employed lipids, the relative amounts of each lipid species in the nanodisc samples determined by $^{31}$P-NMR closely agreed with the amounts we intended to incorporate in the nanodiscs (Supplementary Fig. 2).

To confirm that A$_{2A}$AR[A289C] was folded in nanodiscs across the range of studied lipid compositions, we recorded fluorescence thermal shift assays for A$_{2A}$AR[A289C] in complex with the antagonist ZM241385 and the agonist NECA (5'-N-ethylcarboxamidoadenosine) in

nanodiscs composed of a wide range of different binary mixtures of zwitterionic and anionic lipids. The assay employed a thiol-reactive dye, N-[4-(7-diethylamino-4-methyl-3-coumarinyl)phenyl]maleimide (CPM), which reacts with cysteines that become solvent accessible upon thermal denaturation[30]. As the nanodisc scaffold protein MSP1D1 contains no cysteines, the assay thus observed a direct response of the thermal unfolding of the receptor. In nanodiscs containing mixtures of different relative amounts of POPC and POPS, we observed thermal unfolding curves consistent with well-folded receptors for both complexes with the antagonist ZM241385 (Supplementary Fig. 3a) and complexes with the agonist NECA (Supplementary Fig. 3b). Interestingly, as the molar fraction of POPS increased, the observed thermal melting temperature of A$_{2A}$AR[A289C] also increased by ~10 °C from nanodiscs containing only POPC to nanodiscs containing only POPS (Supplementary Fig. 3a). This overall trend was also observed for A$_{2A}$AR[A289C] in nanodiscs composed of different ratios of POPE and POPS, and POPC with POPA or POPG (Supplementary Fig. 3b), though the trend was most pronounced for nanodiscs containing mixtures of POPC or POPE mixed with POPS. Further, the increase in melting temperature associated with increasing amounts of anionic lipids was observed for both the complex with an antagonist and the complex with an agonist (Supplementary Fig. 3b). To confirm the fluorescence shift assay reported specifically on the thermal unfolding of the receptor within formed nanodiscs rather than disassembly of the nanodiscs, we recorded variable temperature circular dichroism (CD) data of nanodiscs without receptor (Supplementary Fig. 4). These data showed the thermal unfolding of the nanodiscs containing only lipids was >90 °C, the upper limit of our instrument, confirming the fluorescence thermal shift data reported specifically on the unfolding of the receptor. These data appear in line with earlier observations from differential scanning calorimetry measurements, which reported nanodiscs containing binary mixtures of lipids remained assembled up to temperatures from 90 °C to over 105 °C[31].

The pharmacological activity of A$_{2A}$AR[A289C] in lipid nanodiscs containing different binary lipid mixtures was measured in radioligand competition binding assays for both complexes with the antagonist ZM241385 and the agonist NECA. K$_D$ values were determined in nanodiscs containing only POPC lipids and mixtures of POPC with one of three different anionic lipids. Measured K$_D$ values for the antagonist ZM241385 varied only by a factor of ~2 among the different lipid compositions (Supplementary Fig. 5). Measured K$_I$ values for the agonist NECA varied only by a factor of ~3 among different lipid compositions (Supplementary Fig. 5). These relatively small differences indicated A$_{2A}$AR[A289C] is pharmacologically active in the range of lipid compositions studied and, the presence of anionic lipids appeared to not impose a significant difference on the pharmacological activity of A$_{2A}$AR.

### The conformational ensemble of activated A$_{2A}$AR strongly depends on the membrane environment

To prepare samples for NMR studies, a $^{19}$F-NMR reporter group was introduced by reacting the solvent-accessible cysteine at position 289 with 2,2,2-trifluoroethanethiol (TET) using an in-membrane chemical modification approach[32], yielding A$_{2A}$AR[A289C$^{TET}$]. Earlier studies employing both the same stable-isotope labeling methodology and A$_{2A}$AR variant demonstrated no other cysteines were available for $^{19}$F labeling[22]. The A289C $^{19}$F-NMR reporter was highly sensitive to function-related changes in the efficacies of bound drugs, providing 'fingerprints' for the corresponding functional states[22].

$^{19}$F-NMR spectra of A$_{2A}$AR[A289C$^{TET}$] in DDM (n-Dodecyl-β-D-Maltopyranoside)/ CHS (cholesteryl hemisuccinate) mixed micelles with receptor prepared from *Pichia pastoris* (Fig. 1a) were in overall good agreement with $^{19}$F-NMR spectra reported for the same A$_{2A}$AR variant expressed in insect cells and solubilized with the same detergent[22]. The present spectrum of an A$_{2A}$AR[A289C$^{TET}$] complex

with the antagonist ZM241385 contained two signals at $\delta \approx 11.3$ ppm ($P_3$) and $\delta \approx 9.5$ ppm ($P_1$) with the signal $P_3$ being the dominant signal in the spectrum (Fig. 1a). $^{19}$F spectra of the $A_{2A}AR[A289C^{TET}]$ complex with the agonist NECA contained a signal at the chemical shift of $P_1$ and two new signals, $P_2$ and $P_4$, at $\delta \approx 10.7$ ppm and $\delta \approx 13.1$ ppm, respectively, and no signal intensity at the chemical shift for $P_3$ (Fig. 1a).

We prepared samples of $A_{2A}AR[A289C^{TET}]$ in lipid nanodiscs formed with the membrane scaffold protein MSP1D1 containing a mixture of the zwitterionic lipid POPC and anionic lipid POPS at a molar ratio of 70 to 30, respectively. $^{19}$F-NMR spectra of $A_{2A}AR[A289C^{TET}]$ in nanodiscs with this ratio of lipid species showed the same number of signals with highly similar chemical shifts as observed in $^{19}$F-NMR spectra of $A_{2A}AR[A289C^{TET}]$ in detergent micelles for both antagonist-bound and agonist-bound receptors (Fig. 1b). Only a small difference was observed for the chemical shift of state $P_3$ for $A_{2A}AR$ in complex with the antagonist ZM241385, $\delta \approx 10.9$ ppm in lipid nanodiscs versus $\delta \approx 11.3$ ppm in detergent micelles (Fig. 1). Chemical shifts for states $P_1$, $P_2$, and $P_4$ were practically indistinguishable

between $A_{2A}AR[A289C^{TET}]$ in DDM/CHS micelles and the POPC/POPS lipid nanodisc preparation. The highly similar chemical shifts observed for complexes in DDM/CHS mixed micelles and POPC/POPS lipid nanodiscs indicates that the introduced $^{19}$F-NMR probe is responsive to changes in receptor conformation rather than changes in the employed membrane mimetic.

The highly similar chemical shifts between $A_{2A}AR[A289C^{TET}]$ in DDM/CHS micelles and the POPC/POPS lipid nanodisc preparation allowed us to quantitatively compare relative peak intensities between the two different membrane mimetics to investigate the impact of the membrane environment on the fingerprint of $A_{2A}AR$ functional states. For the complex with the antagonist ZM241385, the relative peak intensities for P1 and P3 were only marginally different between $A_{2A}AR$ in detergent and $A_{2A}AR$ in lipid nanodiscs (Fig. 1a). Ratios of peak intensities for apo $A_{2A}AR$ were also highly similar between detergent and lipid nanodisc environments (Supplementary Fig. 6a). In contrast, we observed striking differences in the relative intensities of peaks for the $A_{2A}AR$ complex with agonist. State P4, observed only for agonist-

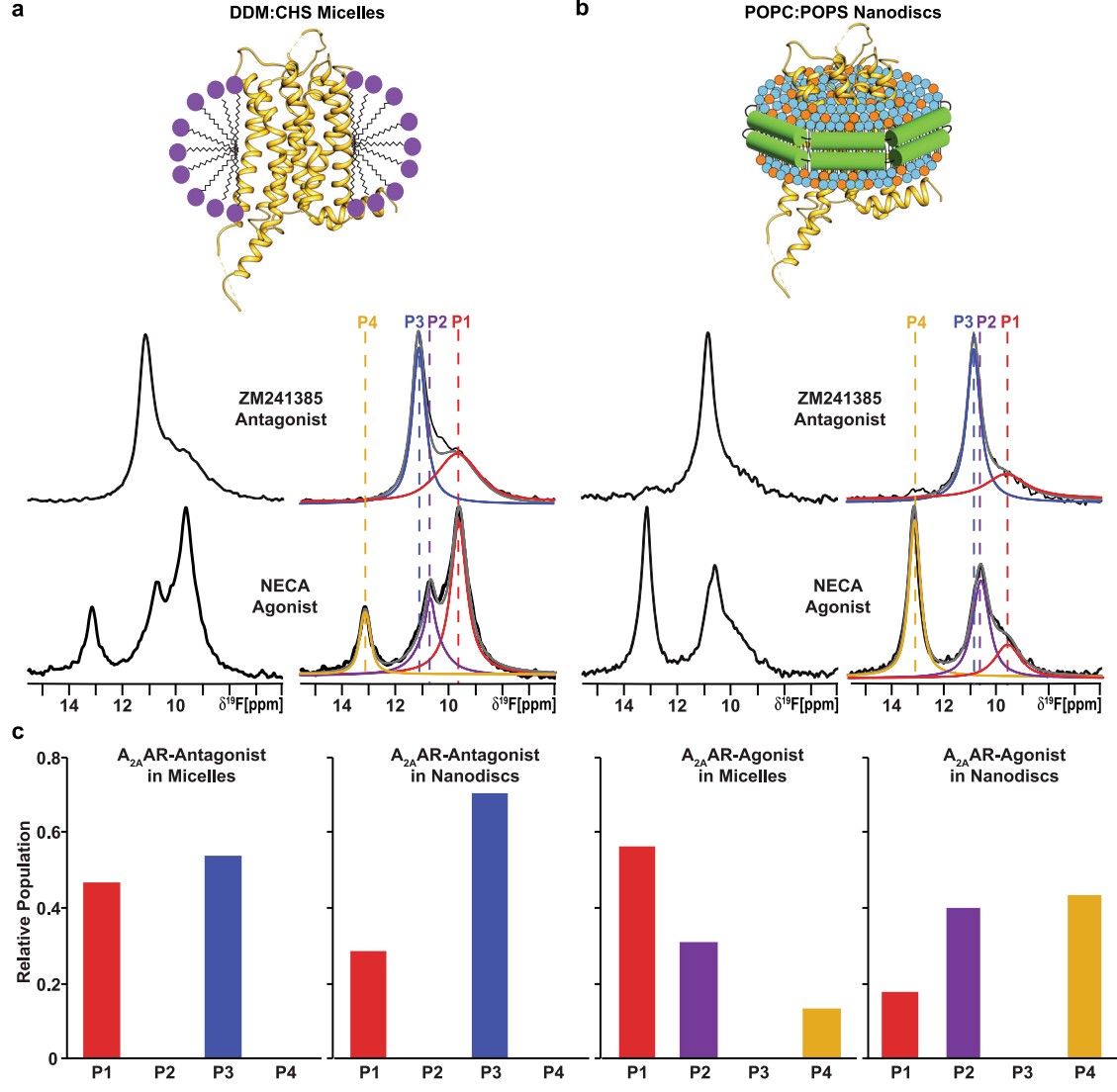

**Fig. 1 | NMR-observed conformational states of $A_{2A}AR$-ligand complexes compared in two different membrane mimetics. a** The 1-dimensional $^{19}$F-NMR spectra of $A_{2A}AR[A289C^{TET}]$ reconstituted into DDM/CHS mixed micelles and in complexes with the antagonist ZM241385 and agonist NECA. On the right, the NMR spectra shown on the left are interpreted by Lorenztian deconvolutions with the minimal number of components that provided a good fit, labeled P1 to P4. The chemical shifts of P1 to P4 are indicated by the colored dashed vertical lines. **b** The 1-dimensional $^{19}$F-NMR spectra of $A_{2A}AR[A289C^{TET}]$ reconstituted into lipid nanodiscs containing the lipids POPC and POPS in a 70:30 molar ratio and in complexes with the same antagonist and agonist. Same presentation details as in **a**. **c** Relative populations of each state represented in a bar chart format. Source data are included as a Source Data file.

bound $A_{2A}AR$, showed nearly a factor of 4 increase in intensity for POPC/POPS nanodiscs (Fig. 1b). State P1, which was the dominant peak observed in DDM/CHS, showed an intensity of nearly a factor of 4 smaller for POPC/POPS nanodiscs (Fig. 1b). The large differences in relative peak intensities were observed only for agonist-bound $A_{2A}AR$, indicating a synergy between drug efficacy and the surrounding membrane environment. The ratio of populations of individual conformers for agonist-bound $A_{2A}AR$ is highly sensitive to changes in the surrounding membrane environment, but not for antagonist-bound $A_{2A}AR$, as quantified in Fig. 1c. We also observed different rates of conformational exchange between $A_{2A}AR$ in DDM/CHS micelles and in lipid nanodiscs in 2-dimensional [$^{19}$F,$^{19}$F]-exchange spectroscopy (EXSY) (Supplementary Fig. 7). In the 2D-EXSY spectrum recorded with agonist-bound $A_{2A}AR$ in POPC/POPS lipid nanodiscs, we did not observe the presence of any crosspeaks. This is in contrast to previously reported EXSY spectra of agonist-bound $A_{2A}AR$ in DDM/CHS micelles that reported exchange crosspeaks between peaks P1 and P2[22], indicating the rate of exchange for populations P1 and P2 must be at least an order of magnitude slower in lipid nanodiscs.

### Agonist-bound $A_{2A}AR$ adopts a predominantly inactive conformation in the absence of anionic lipids

To investigate the impact of anionic lipids on the observed conformational ensemble for $A_{2A}AR$-agonist complexes, we recorded $^{19}$F-NMR data with $A_{2A}AR[A289C^{TET}]$ bound to the agonist NECA in lipid nanodiscs containing only the zwitterionic phospholipid POPC (Fig. 2). Surprisingly, we observed a conformational ensemble that more closely resembled the fingerprint of inactive $A_{2A}AR$ even though a saturating amount of agonist was present, showing a dominant peak P3 and minor populations of peaks P1 and P2 (Fig. 2a). Fluorescence thermal shift assays of the $A_{2A}AR$-agonist complex in POPC showed a well-folded protein (Supplementary Fig. 3), and radioligand binding experiments showed $A_{2A}AR$ in POPC nanodiscs bound the agonist NECA with a measured dissociation constant that was only a factor of ~2.5 times different from nanodiscs containing POPC and POPS (Supplementary Fig. 5). Thus, the $^{19}$F-NMR data show a conformational ensemble of functional, agonist-bound $A_{2A}AR$ largely in an inactive

conformation. This indicates that even in the presence of saturating amounts of an activating drug, $A_{2A}AR$ adopts predominantly an inactive conformation in the absence of anionic lipids. $^{19}$F-NMR data of the $A_{2A}AR[A289C^{TET}]$-agonist complex in POPC nanodiscs measured at the higher temperature of 300 K were highly similar to spectra measured at 280 K with only a minor increase in the population of state P4 (Supplementary Fig. 8).

To test whether $A_{2A}AR$ in POPC nanodiscs could still form signaling complexes, we added to the same sample the engineered $G\alpha_S$ protein, 'mini-$G\alpha_S$', a partner protein designed to induce conformational changes in ternary complexes with $A_{2A}AR$ that mimic those of the native $G\alpha_S$ protein[33,34]. Analytical size exclusion chromatography and SDS-PAGE analysis confirmed the formation of a the $A_{2A}AR$ ternary complex with NECA and mini-$G\alpha_S$ (Supplementary Fig. 9). $^{19}$F-NMR data of $A_{2A}AR[A289C^{TET}]$ in POPC nanodiscs showed a clear response to the addition of mini-$G\alpha_S$ (Fig. 2b). The NMR data appeared qualitatively similar to data measured for agonist-bound $A_{2A}AR$ in nanodiscs containing POPC and POPS, where the population of state P3 was significantly reduced and the populations of states P2 and P4, specific to the agonist complex, increased (Fig. 2b). This data allowed us to unambiguously assign the identity of state P4 to a conformation of activated $A_{2A}AR$ in the ternary complex. However, for the ternary complex, the relative population of state P4 was ~1/3 to 1/4 of the intensity of P4 observed for $A_{2A}AR$-agonist complex in POPC/POPS nanodiscs. This indicated that while $A_{2A}AR$ could still populate a fully active conformation, the much lower P4 signal intensity indicated that the corresponding population was much lower than that observed for $A_{2A}AR$-agonist in the presence of POPS even in the absence of a partner signaling protein (Fig. 1b).

### $A_{2A}AR$ activation depends on lipid headgroup charge rather than chemical scaffold

To investigate whether observations in Fig. 2 were specific to POPS or applied more generally to additional anionic lipids, we prepared a series of agonist-bound $A_{2A}AR$ samples in nanodiscs containing binary mixtures of POPC and a second, different anionic lipid. $^{19}$F-NMR data of agonist-bound $A_{2A}AR$ were all qualitatively highly similar for nanodiscs containing POPC mixed with one type of anionic lipid, including with POPS, POPA, POPG, or PI(4,5)P2 (Fig. 3b). In all mixtures of POPC with anionic lipids, we saw a significant population of the peak P4 corresponding to activated $A_{2A}AR$ (Fig. 3b). To also investigate the possibility that POPC played a special role in the binary mixtures, we recorded NMR data with agonist-bound $A_{2A}AR$ in nanodiscs containing POPE mixed with POPS. NMR data with these samples was highly similar to samples prepared with POPC and POPS, showing the presence of activated $A_{2A}AR$ with a population that increased proportionally with increasing amounts of anionic lipids (Supplementary Fig. 10). These data support that activation of agonist-bound $A_{2A}AR$ could be achieved in the presence of any anionic lipid studied regardless of the phospholipid headgroup chemical structure. We also determined the integrals of each peak from the deconvolutions of all the spectra and tabulated these values as a fraction of the total integrated signal intensities (Supplementary Table 1). For all spectra shown in Fig. 3b we observed comparable integrated intensities for states, with the state P4 showing the largest fraction in the presence of PIP2 (Supplementary Table 1). We also observed larger line widths for states P4 and P2 for the $A_{2A}AR$ agonist complex in nanodiscs containing mixtures of POPC and POPA, suggesting the active state exhibits a larger degree of structural plasticity for this lipid composition.

### Positively charged amino acids at the $A_{2A}AR$ intracellular surface of TM VI direct lipid-induced conformational changes

Our $^{19}$F data showed anionic lipids impacted the conformational ensemble of agonist-bound but not antagonist-bound $A_{2A}AR$. We therefore hypothesized positively charged residues located near the

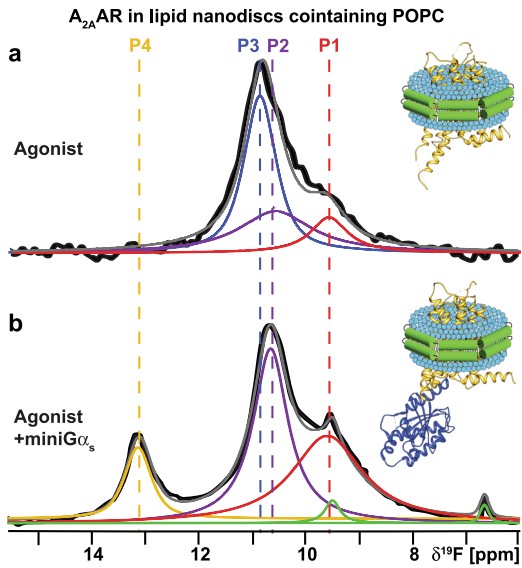

**Fig. 2 | NMR-observed conformation of an $A_{2A}AR$-agonist complex and ternary complex in nanodiscs containing only zwitterionic lipids. a** A 1-dimensional $^{19}$F-NMR spectrum of an $A_{2A}AR[A289C^{TET}]$ complex with the agonist NECA in lipid nanodiscs containing the zwitterionic lipid POPC. **b** The 1-dimensional $^{19}$F-NMR spectrum of an $A_{2A}AR$ ternary complex with the agonist NECA and an engineered $G\alpha_S$ protein in lipid nanodiscs containing POPC. Other figure presentation details the same as in Fig. 1, a and b.

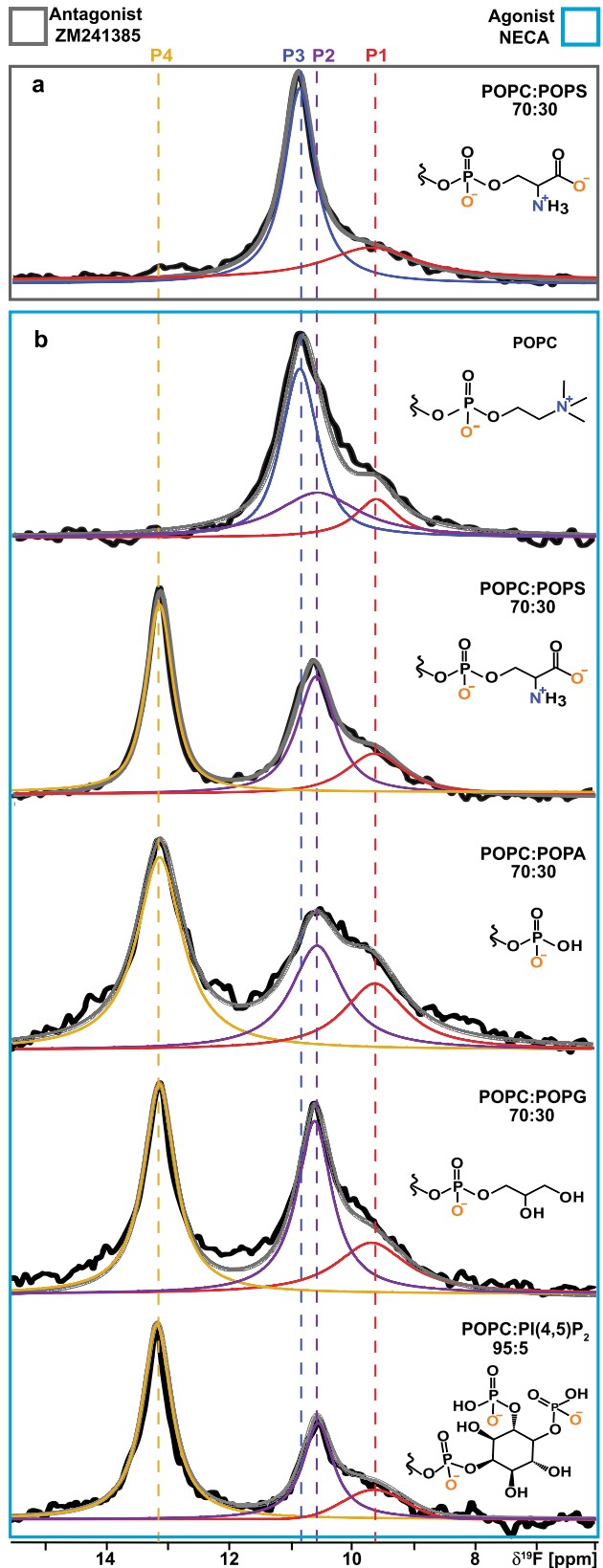

**Fig. 3 | A$_{2A}$AR conformational response to the presence of anionic lipids with varying chemical scaffolds. a** A 1D $^{19}$F-NMR spectrum of the antagonist-A$_{2A}$AR[A289C$^{TET}$] complex in lipid nanodiscs containing a mixture of POPC and POPS lipids. **b** The 1D $^{19}$F NMR spectra of an agonist-A$_{2A}$AR[A289C$^{TET}$] complex in lipid nanodiscs containing either only POPC lipids or a mixture of POPC and anionic lipids, as indicated on the right of each spectrum. The chemical structures of the lipid headgroups are shown with charges expected at physiological pH.

nomenclature[36]) and in different conformations between agonist-bound and antagonist-bound A$_{2A}$AR crystal structures (Fig. 4a). Though our experiments were carried out at neutral pH, H230$^{6.32}$ has been proposed to be positively charged[37] and was thus included in the current study. Positively charged amino acids are frequently found in position 6.32 in other GPCRs (see Discussion).

We prepared A$_{2A}$AR variants by individually replacing each of these four residues with the polar but neutral amino acid glutamine. Each of the resulting variants was folded and demonstrated closely similar pharmacological activity compared with native A$_{2A}$AR (Supplementary Fig. 11). In thermal shift assays, the variants also showed an increase in melting temperature with increasing amounts of anionic lipids, similar to A$_{2A}$AR[A289C] (Supplementary Fig. 11). For two of the variants, A$_{2A}$AR[A289C$^{TET}$,R199Q] and A$_{2A}$AR[A289C$^{TET}$,R291Q], $^{19}$F-NMR spectra for complexes with the agonist NECA were similar to the data of A$_{2A}$AR[A298C$^{TET}$] in both nanodiscs containing POPC or a mixture of 70/30 POPC/POPS (Fig. 4b). For A$_{2A}$AR[A289C$^{TET}$,R291Q], state P4 showed a higher relative intensity in nanodiscs containing 70/30 POPC/POPS but the same overall response to the presence of anionic lipids as A$_{2A}$AR[A289C$^{TET}$].

For the two variants with mutations in helix VI, A$_{2A}$AR[A289C$^{TET}$,H230Q] and A$_{2A}$AR[A289C$^{TET}$,K233Q], we observed a completely different response to the presence of anionic lipids and in striking contrast to A$_{2A}$AR[A289C$^{TET}$]. In the absence of anionic lipids, the $^{19}$F NMR spectra of these two variants in complex with the agonist NECA resembled the spectra of A$_{2A}$AR[A289C$^{TET}$] measured in the presence of anionic lipids (Fig. 4). $^{19}$F spectra of A$_{2A}$AR[A289C$^{TET}$,H230Q] and A$_{2A}$AR[A289C$^{TET}$,K233Q] in complex with NECA and in nanodiscs containing a mixture of POPC and POPS (70:30) appeared similar to spectra of inactive A$_{2A}$AR[A298C$^{TET}$] (Fig. 4). Intriguingly the individual mutations in Helix VI appeared to completely reverse the sensitivity of A$_{2A}$AR to anionic lipids.

We tested signaling complex formation with A$_{2A}$AR[A289C$^{TET}$,K233Q] with mini Gα$_S$ in nanodiscs containing POPC and POPS (70:30). Upon addition of mini-Gα$_S$, $^{19}$F-NMR spectra showed the presence of the state P4 (Supplementary Fig. 12), indicating complex formation can still occur for this variant. Cyclic AMP accumulation assays showed dose-dependent responses to agonists for the four A$_{2A}$AR variants were similar to wild-type A$_{2A}$AR (Supplementary Fig. 13), with A$_{2A}$AR[A289C$^{TET}$,R291Q] showing a measurably higher basal level of activity, consistent with the relatively higher signal intensity for peak P4 in $^{19}$F-NMR data (Fig. 4). Consideration of the $^{19}$F-NMR data and correlative functional experiments together indicates anionic lipids may not be strictly required for A$_{2A}$AR activation but shift the conformational equilibria toward a fully active conformation even in the absence of a partner G protein.

## Anionic lipids prepare A$_{2A}$AR to recognize intracellular G proteins

To further investigate the molecular mechanism by which anionic lipids modulated A$_{2A}$AR activation, a series of all-atom molecular dynamics (MD) simulations were implemented with several different A$_{2A}$AR conformations and with different lipid compositions. We first noted H230$^{6.32}$, K233$^{6.35}$ and R291$^{7.56}$ form a triad that coordinates engagement with mini-Gα$_S$ through interactions with glutamic acid in the end of the α5 helix of mini-Gα$_S$ at position 392 (Fig. 5). The difference between inactive and the fully active G protein-coupled

intracellular surface that show significant differences in conformation between active and inactive states could play important roles mediating the impact of anionic lipids. We identified 4 positively charged residues predicted by the Orientation of Proteins in Membranes (OPM) database[35] to be near the lipid-bilayer boundary: R199$^{5.60}$, H230$^{6.32}$, K233$^{6.35}$ and R291$^{7.56}$ (superscripts denote the Ballesteros-Weinstein

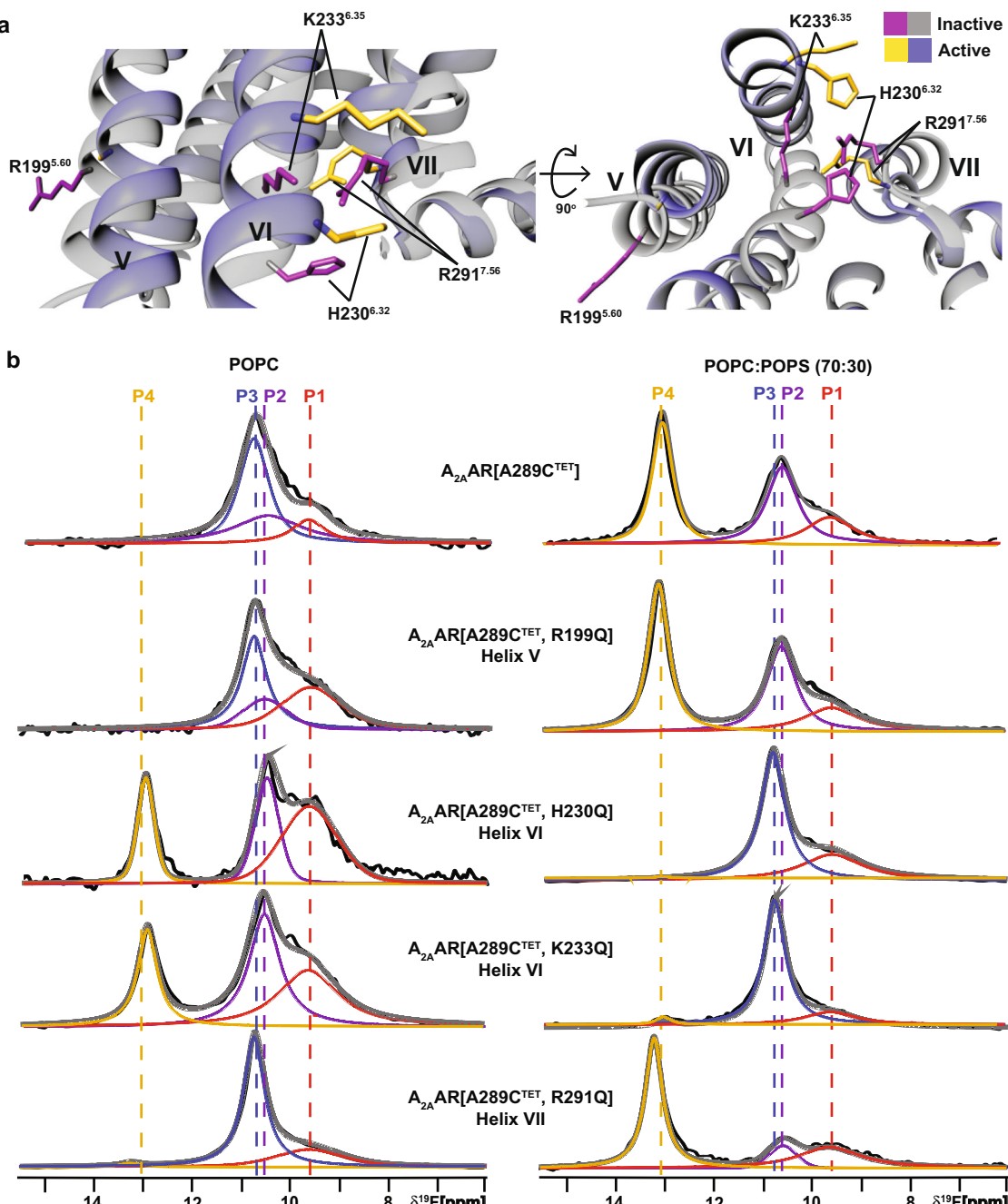

**Fig. 4 | Impact of positively charged amino acids near the intracellular signaling on NMR response to anionic lipids. a** Location of several charged amino acids near the lipid bilayer boundary. **b** $^{19}$F-NMR spectra of A$_{2A}$AR[A289C$^{TET}$] and several A$_{2A}$AR variants in complex with the agonist NECA in lipid nanodiscs containing only POPC lipids (left column), and in complex with the agonist NECA and in lipid nanodiscs containing a mixture of POPC and POPS lipids at a 70:30 molar ratio (right column). Other figure presentation details the same as in Fig. 1, a and b.

states is signaled by the distribution of distances between H230$^{6.32}$ and R291$^{7.56}$. For the A$_{2A}$AR complex with mini-Gα$_S$, the median of this distance distribution shifts toward longer distances and the distribution broadens (Fig. 5 and Supplementary Figs. 14 and 15). We hypothesized that negatively charged headgroups of anionic phospholipids mimic interactions of E392 in mini-Gα$_S$, preorganizing the intracellular face for engaging with the G protein. To test this, we simulated the fully active receptor but without mini-Gα$_S$. Simulations were initialized from the A$_{2A}$AR ternary complex with the agonist NECA and mini-Gα$_S$ (PDB ID 5G53)[33] and then deleting the mini-Gα$_S$ *in-silico* and allowing lipids to access the G protein-binding interface. Simulations were first performed with the backbone of the protein

restrained to keep it close to the fully active receptor. In simulated membranes containing only POPC, the median H230$^{6.32}$ and R291$^{7.56}$ distance for agonist-bound A$_{2A}$AR is similar to that observed for the A$_{2A}$AR inactive state (Fig. 5)—a surprising result, given that the backbone of the protein is kept close to the fully active state by the restraints. In contrast, in the presence of POPS the median H230$^{6.32}$ and R291$^{7.56}$ distance shifts to longer distance, as a PS headgroup intercalates between H230$^{6.32}$ and R291$^{7.56}$ (Fig. 5 bottom right panel). If the restraints are then released and the simulation continued, the median H230$^{6.32}$ and R291$^{7.56}$ distance shifts to still longer distances (Fig. 5b, blue arrow), mimicking the Gα$_S$ protein-bound state. The interaction of the COO- group in PS headgroups with these residues

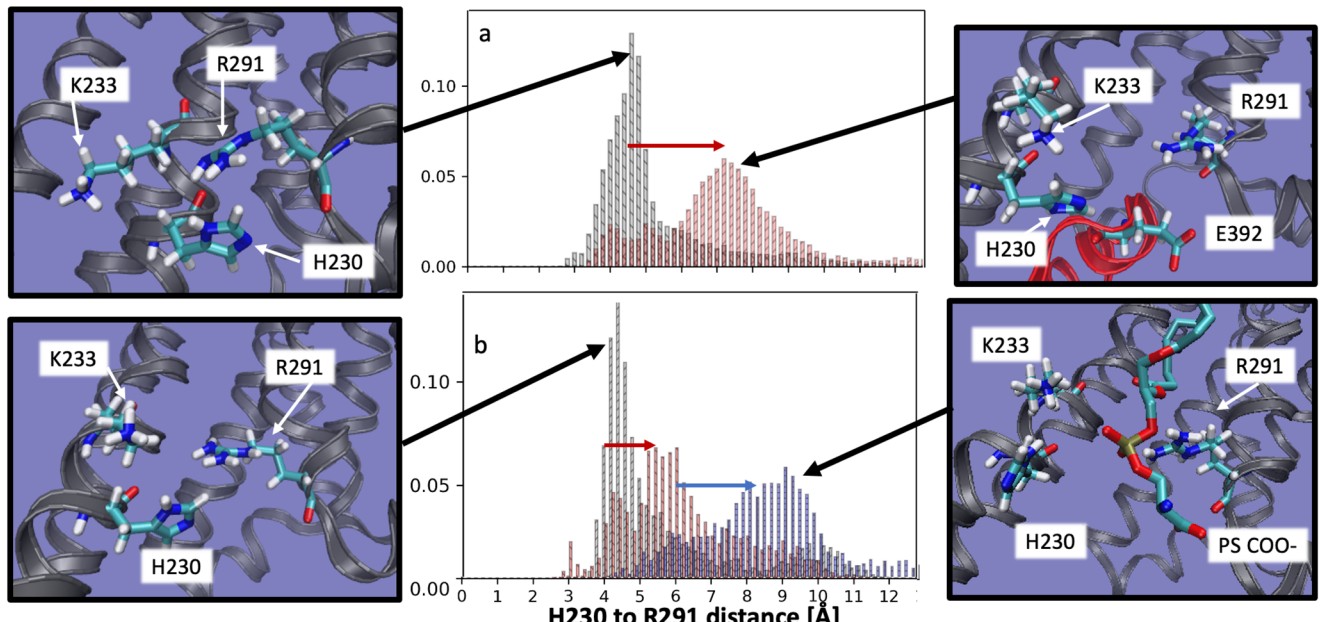

**Fig. 5 | A$_{2A}$AR-lipid interactions observed in molecular dynamics simulations.** Distance distribution between H230$^{6.32}$ and R291$^{7.56}$ shifts to farther distance (red arrow, panel 'a') when the receptor switches from the inactive state (panel 'a', black) to the fully active, G protein-coupled state (panel 'a', red). When simulated in the fully active state but without the G-protein and no POPS (and with the backbone restrained to stay close to the fully active state), the H230$^{6.32}$ to R291$^{7.56}$ distance distribution (panel 'b', black) is similar to the inactive state, but the shift to longer distances is partially recapitulated (red arrow, panel 'b') when simulated in a membrane containing 85% phosphatidyl serine (panel 'b', red). Upon release of the backbone restraints the 85% PS simulation shifts to longer distances (blue arrow, panel 'b'), with a distribution very similar to the fully active, G-protein coupled system. Inspection of the structures representative of the most likely distance for each case reveals that when the H230$^{6.32}$ to R291$^{7.56}$ distance is shorter, reflecting an inactive-like state, H230$^{6.32}$, K233$^{6.35}$ R291$^{7.56}$ form a tight cluster (far left panels, top and bottom). When the distance is longer, reflecting an active-like state, this cluster is split, and coordinates a glutamic acid from the α5 helix of the G-protein (top right panel), or a PS headgroup in the absence of G-protein (lower right panel).

mimics the interactions of the COO- group on E392, but only when the receptor is in the fully active state (Fig. 5).

## Discussion

Results presented in Figs. 3 and 4 are intriguing in the context of literature data on the influence of phospholipids on GPCR activity. In earlier spectrophotometric studies of rhodopsin, anionic lipids shifted the equilibrium of intermediate activation states[38]. In studies of β$_2$AR signaling, anionic lipids impacted the preference of the receptor to interact with Gα$_i$ over Gα$_s$ through lipid-G protein charge complementarity[14]. IC$_{50}$ values for β$_2$AR ligands varied among different lipid compositions in nanodiscs by a factor of ~3 for antagonists and ~7 for agonists[11], though no clear relationship was observed between lipid headgroup type and measured IC$_{50}$ values. In the present work, we observed a variation in K$_D$ values of the antagonist ZM241385 and agonist NECA among different lipid compositions by a factor of ~2 and ~3, respectively, also with no obvious correlation between lipid headgroup and determined K$_D$ value (Supplementary Fig. 5). Our data are more in line with earlier observations of the neurotensin 1 receptor, which reported similar binding affinities of the agonist neurotensin in nanodiscs containing either POPC or mixtures of POPC and POPG[39]. As all NMR samples employed a saturating amount of ligand, the relatively small differences among K$_D$ values do not explain the striking differences observed in $^{19}$F spectra observed between nanodisc preparations containing only zwitterionic lipids versus a mixture of zwitterionic and anionic lipids (Fig. 3). Thus, observed differences in our NMR data between different lipid compositions must be due to lipid-dependent changes in the A$_{2A}$AR conformational equilibria.

Comparing spectra from Figs. 1 and 2, anionic lipids appear to not be strictly required for A$_{2A}$AR complex formation with mini-Gα$_S$, rather they shift the conformational equilibria to favor a fully active A$_{2A}$AR conformer. $^{19}$F-NMR data of agonist-bound A$_{2A}$AR[A289C,K233Q] show this variant can still form signaling complexes even though an active state P4 is not observed in nanodiscs containing anionic lipids. These data are consistent with the cAMP functional data (Supplementary Fig. 13) and support a model whereby anionic lipids control the mechanism of A$_{2A}$AR-G protein recognition (Fig. 6). In the absence of anionic lipids, A$_{2A}$AR can form signaling complexes through an induced fit mechanism. In the presence of anionic lipids, a fully active receptor conformation is populated, leading to a conformational selection process directing complex formation (Fig. 6). Thus, anionic lipids appear to "prime" A$_{2A}$AR for complex formation with partner proteins. This priming effect is reminiscent of earlier studies that proposed G proteins could synergistically prime a receptor for coupling with other G protein subtypes[40].

In earlier $^{19}$F-NMR studies[22] and in the present study, the presence of state P4 is also observed for agonist-bound A$_{2A}$AR in mixed micelles composed of DDM and CHS (Fig. 1). Since the pK$_A$ of the carboxylic acid group of CHS is ~5.8, at the pH value of 7 used for samples in the current study we anticipate the vast majority of CHS molecules should be negatively charged, consistent with earlier experiments with CHS[41]. We therefore hypothesize that the negative charge on CHS may mimic the impact observed on the conformational equilibria of A$_{2A}$AR by the negative charges for anionic lipids.

Our data and proposed mechanism provide an opportunity to evaluate experimental results in the context of observations from computational modeling. MD simulations of A$_{2A}$AR in complex with mini-Gα$_S$ indicated PIP2 strengthened interactions within the complex by "bridging" between basic residues of A$_{2A}$AR and mini-Gα$_S$[27]. $^{19}$F-NMR spectra of A$_{2A}$AR in nanodiscs containing 5% PIP2 and in the absence of partner G protein show the presence of fully active A$_{2A}$AR (Fig. 3), suggesting PIP2 may also enhance complex formation by populating a fully active receptor conformation prior to complex formation and play multiple roles in stabilizing the active complex. In MD simulations

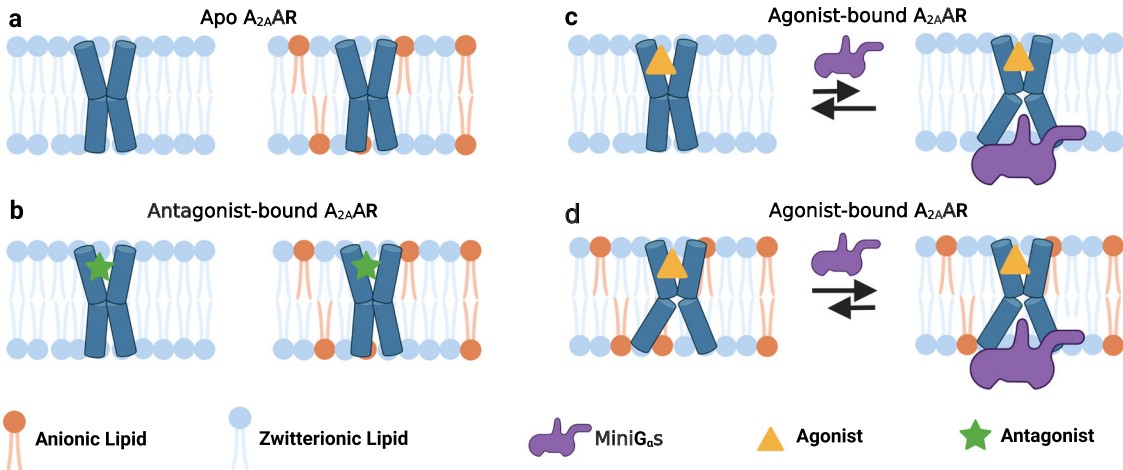

**Fig. 6 | Visualization of the role of anionic lipids in complex formation of $A_{2A}AR$ with $G\alpha_S$. a, b** Schematic side views of **a** apo and **b** antagonist-bound $A_{2A}AR$ in phospholipid membranes without and with anionic lipids, colored blue and orange, respectively. **c, d** Schematic view of agonist-bound $A_{2A}AR$ and mini-$G\alpha_S$ in membranes containing **c** only zwitterionic phospholipids and **d** zwitterionic and anionic phospholipids. The black arrows signify an equilibrium between mini-$G\alpha_S$ bound $A_{2A}AR$ and $A_{2A}AR$ bound to agonist alone, and the length of the arrows signifies how anionic lipids shift this equilibrium toward formation of the ternary complex. This figure was prepared with Biorender.com.

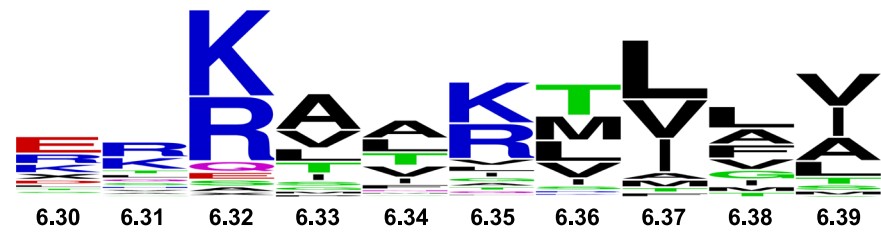

**Fig. 7 | Conservation of residues at the intracellular end of helix 6 among 290 class A GPCRs.** The size of each one-letter amino acid corresponds to the frequency of occurrence for that amino acid type. Numbers indicate the residue position in the Ballesteros-Weinstein nomenclature[36]. Sequences were obtained and aligned in the GPCRdb[43]. The figure was created using the online tool available at https://weblogo.berkeley.edu/logo.cgi.

of $\beta_2AR$[38] and $A_{2A}AR$[37], active conformations of the receptors were proposed to be stabilized by increasing the receptor affinity for agonists and interactions with membrane-facing residues in TM6 and TM7. In contrast, experimentally we observed only relatively smaller differences in agonist binding among different lipid compositions, and we also propose that penetration of anionic lipids into the G protein binding site is important to stabilizing an active $A_{2A}AR$ conformation (Fig. 5).

Conservation of basic residues in helix VI among class A GPCRs and GPCRs from all classes suggest the present observations may extend to many additional receptors. Using structure-based alignment tools in GPCRdb[42,43], we compared sequences of GPCRs among class A and among all classes. In class A receptors, one of these three residues is observed at position 6.32 or 6.35 with frequencies of 75.5% and 59.3%, respectively (Fig. 7 and Table 1). Expanding this search to include 1 position before or after these positions increases the frequency to 88.3% and 64.5%. In the opioid receptor family, position 6.32 is an arginine and has been noted to be critical for the activation of Gi signaling by the mu opioid receptor[44]. Conservation of a positively charged amino acid is also observed among the 402 non-olfactory GPCRs from all families. Notably, for class F GPCRs, either an arginine or lysine is always found in position 6.32 and mutations to a non-positively charged amino acid destroy G protein signaling[45]. In class B receptors, position 6.35 is typically a lysine and noted to be critical for receptor activation[46]. Our study suggests that lipid-receptor interactions may play a critical role in regulating the drug response and activity of GPCRs across all receptor families.

## Methods

### Molecular cloning

The gene encoding human $A_{2A}AR$ (1-316) was cloned into a pPIC9K vector (Invitrogen) at the BamHI and NotI restriction sites. The gene contained a single amino acid replacement (N154Q) to remove the only glycosylation site in the receptor, an N-terminal FLAG tag, and a 10 X C-terminal His tag. We used PCR-based site-directed mutagenesis with the Accuprime *Pfx* SuperMix (ThermoFisher Scientific, Catalog Number: 12344040) to replace A289[7.54] with cysteine, creating $A_{2A}AR$[A289C]. This plasmid was used as a template for generating the additional $A_{2A}AR$ variants R291[7.56]Q, K233[6.35]Q, H230[6.32]Q and R199[5.60]Q via the same site directed mutagenesis approach using nucleic acid oligomers listed in Supplementary Table 2.

### Small-scale protein expression optimization

Plasmids containing $A_{2A}AR$ were introduced into a BG12 strain of *Pichia pastoris* (Biogrammatics, SKU: PS10012) via electroporation. Clones exhibiting high protein expression were identified using small-scale protein production and screening approaches as previously reported[21]. Glycerol stocks of highly expressing clones were stored at −80 °C for future use.

### $A_{2A}AR$ expression, purification, and ¹⁹F-labeling via chemical modification

All $A_{2A}AR$ variants were expressed in *P. pastoris* following previously reported protocols[21]. Briefly, 4 mL cultures in buffered minimal glycerol (BMGY) media were inoculated from glycerol stocks and allowed to grow at 30 °C for 48 h at 200 RPM. These cultures were used to

**Table 1 | Sequence conservation of positively charged amino acids on the intracellular ends of helices V, VI, and VII**

| Residue position | Amino acid type in A$_{2A}$AR | Occurrence frequency of amino acid types K, R, or H (%) | | | |
|---|---|---|---|---|---|
| | | 290 class A GPCRs | 290 class A GPCRs within ±1 residue | 402 non-olfactory GPCRs | 402 non-olfactory GPCRs ±1 residue |
| 5.60 | R | 39.6% | 41.03% | 34.57% | 40.80% |
| 6.32 | H | 75.5% | 88.27% | 60.19% | 70.90% |
| 6.35 | K | 59.3% | 64.48% | 45.77% | 64.94% |
| 7.56 | R | 9.3% | 11.03% | 8.46% | 14.18% |

inoculate 50 mL BMGY medium allowed to grow at 30 °C for 60 h with 200 RPM shaking. Each 50 mL culture was subsequently used to inoculate 500 mL BMGY medium and allowed to incubate at 30 °C for 48 h and 200 RPM. Cultures were then centrifuged at 3000 x g for 15 min, the supernatant discarded, and then resuspended in 500 mL of buffered minimal methanol (BMMY) medium without methanol. Cultures were allowed to grow for 6 h at 28 °C to remove any remaining glycerol. Protein expression was induced by the addition of methanol to a final concentration of 0.5% w/v. Two further aliquots of 0.5% w/v methanol were added to the cultures at 12 h intervals after induction for a total expression time of 36 h. Cells were harvested by centrifugation at 3000 x g for 15 min. Cell pellets were frozen in liquid nitrogen and stored at −80 °C until needed. Cell pellets were resuspended and lysed in lysis buffer (50 mM sodium phosphate pH 7.0, 100 mM NaCl, 5% glycerol (w/v), and in-house prepared protease inhibitor cocktail solution). Cell membranes containing A$_{2A}$AR were isolated and collected by ultracentrifugation at 200,000 x g for 30 min, frozen in liquid nitrogen, and stored at −80 °C for future use.

[19]F-TET was introduced to A$_{2A}$AR variants using the in-membrane chemical modification (IMCM) method as previously reported[32]. In brief, isolated membrane pellets were resuspended in buffer (10 mM HEPES pH 7.0, 10 mM KCl, 20 mM MgCl$_2$, 1 M NaCl, 4 mM theophylline) and incubated with 1 mM of 4,4'-dithiodipyridine (aldrithiol-4) and protease inhibitor cocktail solution (prepared in-house) for 1 h at 4 °C. The suspension was pelleted using ultracentrifugation, and excess aldrithiol was washed off using the same buffer. Membranes were resuspended and incubated with 1 mM of 2,2,2-trifluoroethaethiol (TET) for 1 h at 4 °C. The suspension was pelleted using ultracentrifugation and the resulting pellet washed with buffer. Membranes were resuspended in the same buffer and incubated with 1 mM theophylline and protease inhibitor cocktail solution (prepared in-house) for 30 min at 4 °C. The membrane suspension was mixed 1:1 with a solubilizing buffer (50 mM HEPES pH 7.0, 500 mM NaCl, 0.5% (w/v) n-Dodecyl-β-D-Maltopyranoside (DDM), and 0.05% cholesteryl hemisuccinate (CHS)) for 6 h at 4 °C. Insolubilized material was separated by ultracentrifugation at 200,000 x g for 30 min, and the supernatant was incubated overnight with Co$^{2+}$-charged affinity resin (Talon, Clontech, Catalog Number: NC9306569) and 30 mM imidazole at 4 °C.

After overnight incubation, the Co$^{2+}$-resin was washed with 20 column volumes (CV) of wash buffer 1 (50 mM HEPES pH 7.0, 500 mM NaCl, 10 mM MgCl$_2$, 30 mM imidazole, 8 mM ATP, 0.05% DDM, and 0.005% CHS), and washed 2 subsequent times with 20 CV of wash buffer 2 (25 mM HEPES pH 7.0, 250 mM NaCl, 5% glycerol, 30 mM imidazole, 0.05% DDM, 0.005% CHS, and an excess of ligand). A$_{2A}$AR was eluted with buffer 3 (50 mM HEPES pH 7.0, 250 mM NaCl, 5% glycerol, 300 mM imidazole, 0.05% DDM, 0.005% CHS, and ligand). The eluted protein was exchanged into buffer (25 mM HEPES pH 7.0, 75 mM NaCl, 0.05% DDM, 0.005% CHS, 100 μM trifluoroacetic acid (TFA), and ligand) using a PD-10 desalting column (Cytiva, Catalog Number: 17085101) and stored at 4 °C for nanodisc assembly. All buffers prepared with ligands used a saturating concentration of the required ligand. Apo A$_{2A}$AR was prepared without ligand added in the purification buffers.

## Nanodisc assembly

Assembly of lipid nanodiscs containing A$_{2A}$AR followed protocols from previous studies[47–49] optimized to facilitate samples prepared with a wider range of lipid compositions. The scaffold protein MSP1D1 was expressed and purified similarly as described in previous studies[47,48]. 100 mM stock solutions of all lipids were prepared in a cholate buffer (25 mM Tris-HCl, pH 8.0, 150 mM NaCl, and 200 mM sodium cholate). To initiate nanodisc assembly, 27 μM of purified A$_{2A}$AR in DDM/CHS micelles was mixed with purified MSP1D1 and detergent-solubilized lipids in a molar ratio of 1:5:250, respectively. The mixture was incubated for 1-2 h at 4 °C and incubated overnight with pre-washed bio-beads (Bio-Rad Laboratories, Catalog Number: 1528920) at 4 °C. Following the overnight incubation, the bio-beads were removed, and the resulting mixture was incubated with Ni-NTA resin (GoldBio, Catalog Number: H3505) for 24 h at 4 °C. The Ni-resin was collected after 24 h and washed with 2 CV of a wash buffer (50 mM HEPES, pH 7.0, 150 mM NaCl, and 10 mM imidazole). Nanodiscs containing A$_{2A}$AR were eluted with buffer (50 mM HEPES, pH 7.0, 150 mM NaCl, 300 mM imidazole and ligand) and exchanged into a final buffer used for all experiments (25 mM HEPES pH 7.0, 75 mM NaCl, 100 μM TFA and ligand) using a PD-10 desalting column (Cytiva). All ligand-containing buffers were prepared with a saturating concentration of ligand.

## Mini-Gα$_s$ expression and purification

The protocol for mini-Gα$_s$ expression and purification was adapted from previously described studies[33,34]. The purified protein was exchanged into storage buffer (10 mM HEPES pH 7.5, 100 mM NaCl, 10% v/v glycerol, 1 mM MgCl$_2$, 1 μM GDP, 0.1 mM TCEP), concentrated to 1 mM, aliquoted, frozen in liquid nitrogen and stored at −80 °C for future use.

## A$_{2A}$AR-Mini-Gα$_s$ complex formation

Assembly of A$_{2A}$AR ternary complexes with agonist and mini-Gα$_S$ followed a protocol adapted from previous studies[33,34]. Nanodiscs containing A$_{2A}$AR were mixed with a 2-fold molar excess of mini-Gα$_s$, 1 mM MgCl$_2$ and apyrase (0.1 U), and the mixture was incubated at 4 °C overnight. The samples were purified by SEC using a Superdex 200 Increase 10/300 GL column (Cytiva, Catalog Number: 2899094) pre-equilibrated with SEC buffer (25 mM HEPES pH 7.0, 75 mM NaCl, 5 mM MgCl$_2$). Peak fractions, containing the A$_{2A}$AR−mini-Gα$_s$ complex, were pooled and concentrated to 180-200 μM for NMR experiments.

## NMR spectroscopy

Nanodisc samples were concentrated to ~200 μM in 280 μL in a Vivaspin-6 concentrator with a 30 kDa MWCO (Sartorius, Catalog Number: VS0621). 20 μL D$_2$O was added and gently mixed into the sample. [19]F-NMR and [31]P-NMR experiments were measured on a Bruker Avance III HD spectrometer operating at 600 MHz [1]H nutation frequency using Topspin 3.6.2 and equipped with a Bruker 5-mm BBFO probe. To make direct comparisons with previously published [19]F-NMR data[50], [19]F-NMR spectra were measured at 280 K. [31]P-NMR experiments were measured at 300 K to obtain improved spectral resolution. Temperatures were calibrated from a standard sample of 4% methanol in D$_4$-MeOH.

1-dimensional $^{19}$F data were recorded with a data size of 32k complex points, an acquisition period of 360 ms, 16k scans, 120 μs dwell time, and 0.3 s recycle delay for a total experimental time of about 3 h per experiment. All $^{31}$P NMR experiments were acquired with an acquisition time of 900 ms, 2k scans, and 0.3 s recycle delay for a total experiment time of 42 min per experiment.

2-dimensional [$^{19}$F,$^{19}$F]-EXSY experiments were recorded with a data size of 120 and 8192 complex points in the indirect and direct dimensions, respectively. We recorded 256 scans for each experiment with 100 ms of mixing time.

### Radioligand binding assays

Competition binding assays with nanodiscs containing A$_{2A}$AR were recorded as previously described[28]. Ligand binding was measured with 0.125-0.25 μg nanodiscs containing A$_{2A}$AR per sample incubated in buffer containing 25 mM HEPES pH 7.0, 75 mM NaCl, [$^3$H]ZM241385 (American radiolabeled chemicals, SKU: ART 0884-50 μCi) and increasing amounts of ZM241385 or NECA for 60 min at 25 °C. The binding reaction was terminated by filtration with a Microbeta filtermat-96 cell harvester (PerkinElmer). Radioactivity was counted using a MicroBeta2 microplate counter (PerkinElmer). ZM241385 and NECA binding affinities (K$_D$ or K$_I$) were determined using competition binding experiments. Specific binding of A$_{2A}$AR and A$_{2A}$AR variants were determined as the difference in binding obtained in the absence and presence of 10 μM ZM241385. Radioligand experiments were conducted in triplicate and IC$_{50}$ values determined using a nonlinear, least-square regression analysis (Prism 8; GraphPad Software, Inc.). The IC$_{50}$ values were converted to K$_I$ values using the Cheng–Prusoff equation[51]. Error bars for each sample were calculated as the standard error of mean (s.e.m) for $n = 3$ independent experiments.

### Fluorescent thermal shift experiments

Fluorescent thermal shift experiments followed a protocol adapted from earlier publications[30,52]. For each sample, 10 μg of nanodisc sample was incubated in buffer (50 mM HEPES pH 7.0, 150 mM NaCl) containing N-[4-(7-diethylamino-4-methyl-3-coumarinyl)phenyl] maleimide (CPM; Invitrogen, Catalog Number: D346) at a final concentration of 10 μM and incubated for 30 min in the dark on ice. Fluorescence thermal shift experiments were carried out with a Cary Eclipse spectrofluorometer using quartz cuvettes (Starna Cells, Inc.) over a linear temperature range from 20 °C to 90 °C at a constant heating rate of 2 °C/min. The excitation and emission wavelengths were 387 nm and 463 nm, respectively. Fluorescence thermal shift data were analyzed in Origin 8.5 (OriginLab Corporation). The raw data were fit to a Boltzmann sigmoidal curve to determine the melting temperature (T$_m$). Error bars for each sample were calculated as the standard error of mean (s.e.m) for n ≥ 3 independent experiments.

### NMR data processing and analysis

All NMR data were processed and analyzed in Topspin 4.0.8 (Bruker Biospin). All 1-dimensional $^{19}$F-NMR data were processed identically. Prior to Fourier transformation, the spectra were zero-filled to 64k points and multiplied by an exponential window function with 40 Hz line broadening. $^{19}$F spectra were referenced to the TFA signal at −75.8 ppm, which was set to 0 ppm. Deconvolution of the $^{19}$F-NMR data followed previously published procedures[50] and was done with MestreNova version 14.1.1-24571 (MestreLab Research S.L.). The $^{19}$F-NMR spectra were fit with a double- or triple-Lorentzian function. The quality of the fits was assessed from the residual difference between the experimental data and the sum of the computed components.

All $^{31}$P NMR data were processed identically. Prior to Fourier transformation, $^{31}$P spectra were zero-filled to 64k points and multiplied by an exponential window function with 50 Hz line broadening.

The 2-dimensional [$^{19}$F, $^{19}$F] EXSY experiments were processed by zero-filling to 1k (t$_1$) * 2k (t$_2$) points and 100 Hz of exponential line broadening was applied prior to Fourier transformation.

### Computational simulations

Three different A$_{2A}$AR conformations were used as starting configurations: the complex with the antagonist ZM241385 (PDB ID: 3EML)[53], the ternary complex with the full agonist NECA and an engineered mini-G protein (PDB ID: 5G53)[33], and a conformation of the same ternary complex with the mini-G protein deleted. For the latter simulation, production simulations were run with the backbone heavy atoms softly restrained to stay close to the fully active state by harmonic restraints with a 0.5 kJ/mol/Å$^2$ spring constant. The T4-lysozyme was removed from the inactive structure and initial coordinates for residues 209-219 were obtained from the structure of a thermostabilized A$_{2A}$AR variant mutant (PDB: 3PWH)[54] for which those residues are resolved. The N- and C-termini were capped with the default chemistry in the CHARMM force-field, a primary amine at the N-terminus and a carboxylate at the C-terminus.

All simulations were performed with Gromacs 2020.4. Each system was prepared individually for production simulation using the CHARMMGUI membrane builder tools[55]. The protein was embedded in a mixture of POPS and POPC (Supplementary Table 3), solvated with TIP3P water[56], neutralized with Na+ ions if needed, and NaCl added to bring the ionic strength to 150 mM. Lipids and protein were modeled with the CHARMM36 force field[57,58]; ligands were modeled with CHARMM general force field[59]. Final system sizes were at least 15 nm in each dimension, yielding lipid-to-protein ratios of ~750:1.

Each system was prepared individually for production simulation through a series of 6 minimization and heating steps: (i) steepest descent to minimize the initial configuration; (ii) 125,000 steps of leapfrog dynamics with a 1 fsec timestep and velocities reassigned every 500 steps; (iii) 125,000 steps of leapfrog dynamics with a 1 fsec timestep, pressure controlled by the Parinello-Rahman barostat[60] and velocities reassigned every 500 steps, then a total of 750,000 steps of leapfrog dynamics with a 2 fsec timestep and hydrogen positions constrained by LINCS[61], pressure controlled by the Parinello-Rahman barostat, and velocities reassigned every 500 steps. During equilibration, double bonds were restrained in the *cis* configuration to prevent isomerization; these restraints are gradually reduced during the final three stages of the equilibration protocol. Production simulations (NPT ensemble) were integrated with leapfrog using the Parinello-Rahman barostat to control pressure (time constant 5 psec; compressibility 4.5e$^{-5}$ bar$^{-1}$; coupled anisotropically to allow independent fluctuation of the in-plane and normal directions) and temperature controlled using Nose-Hoover[62] (time constant 1 psec) at a temperature of 25 °C. Hydrogens were constrained with LINCS (expansion order 4), a 2 fsec timestep was used, short range electrostatics were computed directly within 1.2 nm, and long-range electrostatics were computed every timestep using particle mesh Ewald[63] with a grid spacing of 1 Å and cubic interpolation. Long range dispersion was smoothly truncated over 10-12 nm using a force-switch cutoff scheme. Residue-residue distances as reported in Fig. 5 were measured between the closest sidechain nitrogen atoms in H230$^{6.32}$ and R291$^{7.56}$.

### cAMP accumulation assays

Plasmids encoding wild-type or variant human A$_{2A}$ adenosine receptors were transfected into CHO-K1 cells (ATCC product CCL-61) using lipofectamine 2000. The cell line was authenticated by the manufacturer and also determined to be free from mycoplasma via a PCR-based assay, agar culture method, and Hoechst DNA stain method. 24 h after transfection, cells were detached and grown in 96-well plates in medium containing equal volume of DMEM and F12 supplemented with 10% fetal bovine serum, 100 Units/ml penicillin, 100 μg/ml streptomycin, and 2 μmol/ml glutamine. After growing for 24 h, culture

medium was removed and cells were washed twice with PBS. Cells were then treated with assay buffer containing rolipram (10 μM) and adenosine deaminase (3 units/ml) for 30 min followed by addition of agonist and incubated for 20 min. The reaction was terminated upon removal of the supernatant, and addition of 100 μl Tween-20 (0.3%). Intracellular cAMP levels were measured with an ALPHAScreen cAMP assay kit (PerkinElmer, Catalog Number: 6760635D) following the manufacture's protocol.

### Reporting summary

Further information on research design is available in the Nature Portfolio Reporting Summary linked to this article.

## Data availability

NMR source data are made available through the Open Science Framework [https://doi.org/10.17605/OSF.IO/56HCU] accessed via the following permanent link https://osf.io/56hcu/?view_only=6f99d22cb0e8439fbb59d2bfa47739f4. Previously published structures from the PDB can be accessed via accession codes 3EML, 5G53 and 3PWH. Simulation input parameters, initial coordinates and final coordinates are made available through the Open Science Framework [https://doi.org/10.17605/OSF.IO/56HCU] accessed via the following permanent link [https://osf.io/56hcu/?view_only=6f99d22cb0e8439fbb59d2bfa47739f4]. Additional source data are provided with this paper as a Source Data file.

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

## Acknowledgements

This work is supported by the National Institutes of Health, NIGMS MIRA grant R35GM138291 (M.T.E.) and by the University of Florida College of Liberal Arts and Sciences. A portion of this work was supported by the McKnight Brain Institute at the National High Magnetic Field Laboratory's AMRIS Facility, which is funded by National Science Foundation Cooperative Agreement No. DMR-1644779 and the State of Florida. The authors also acknowledge support from the NIH/NIDDK Intramural Research Program (ZIA DK031117). L.S. and E.L. were supported by NIH award RO1GM120351. This work used the Extreme Science and Engineering Discovery Environment (XSEDE), which is supported by National Science Foundation grant number ACI-1548562.

## Author contributions

N.T. and A.P.R. expressed A$_{2A}$AR, prepared samples, recorded NMR data, recorded fluorescence thermal shift data and analyzed NMR data with input from M.T.E. A.D. and A.V.W. expressed A$_{2A}$AR and A$_{2A}$AR variants and prepared lipid nanodisc samples. B.J. and N.G.P. expressed A$_{2A}$AR variants, prepared lipid nanodisc samples, and collected NMR data. S.O. recorded ligand binding data with A.P.R. and C.R.M. Z.-G.G. and K.A.J. recorded and analyzed cAMP signaling data. E.L. and L.S. performed computational modeling, simulations and analysis of simulations. M.T.E. designed the study, recorded NMR data, analyzed NMR and other biophysical data and wrote the manuscript with input from all authors.

## Competing interests

The authors declare no competing interests.
