## [Peer Review File · Nature Communications]

Anionic Phospholipids Control Mechanisms of GPCR-G Protein RecognitionReviewers' Comments:

Reviewer #1:

Remarks to the Author:

The article "Anionic Phospholipids Control Mechanisms of GPCR-G Protein Recognition" by Thakur et al. investigates the influence of anionic lipids on various receptor states. The authors find a strong influence of membrane composition (especially anionic lipids) on the dynamic ensemble of A2AAR conformations. While the part of the manuscript describing the experimental data seems well elaborated, the MD simulations do not reach the necessary quality. In general, there is a lack of detailed presentation of the result data from the MD simulations. Some parts are even misleading. Therefore, in the current form, it is difficult to assess the validity of the simulations performed.

Major issues:

- (1) The authors should provide information on the setup (and length) of all MD-simulations. The reference in line 630 "see Table SXX for simulated systems" does lead to nowhere. What was the exact composition of the simulation (e.g. PC/PS ratio or L/P ratio)?
- (2) The structure of the activate receptor state (in absence of the G-protein) was restrained during the production runs. Nevertheless, distance data were extracted from these simulations even without mentioning the restraints in the main text. This is at least misleading if not a major problem.
- (3) This lack of information reduces the impact of the otherwise potentially relevant observation that PS headgroups make contact with the H230/K233/R291 cluster. Is there a way to prove this observation experimentally, e.g. by NMR?
- (4) lines 355-357: The authors state the following: "In the absence of anionic lipids, A2AAR can form signaling complexes through an induced fit mechanism." A similar statement is also made in the abstract. In my opinion, this very strong statement is not fully supported by the available data. I agree that the mechanism of induced matching is relatively likely in this scenario, but why shouldn't conformational selection also be possible? It could simply be that in the absence of anionic lipids, the "correct" conformation for G protein binding is much less likely than in the presence of anionic lipids. For example, in Fig. 5b, the histograms clearly overlap, and all distances appear for both cases, just with different probabilities. Are there other data to distinguish between induced matching and conformational selection in the absence of anionic lipids?

More issues:

- (5) The authors report (from line 306): 'We noted H230, K233 and R291 form a triad that coordinates engagement with mini-Gas'. Does this interaction already exist in the underlying structure? Has it already been reported?
- (6) From line 309: 'The difference between inactive and fully-active G protein coupled states is signaled by the distribution of distances between H230 and R291.' Is this just a claim of the authors? What is the rationale for this claim? In practice, the distance between TM6 and TM3/TM4 is more likely to be used to monitor the activation state of the receptor, as it reflects the outward tilt of TM6.
- (7) In the manuscript, the authors use the term "tertiary complex", which seems rather unusual to me. I would recommend to use "ternary complex" instead.
- (8) Line 442: I am not sure if a distance between two residues can be 'active-like' or 'inactive-like', but it could refer to such a receptor state.
- (9) Was the T4 lysozyme removed for the simulations? If so, how was the resulting gap in the structure handled?
- (10) How were the protein termini capped?
- (11) What were the protonation states of amino acids crucial for receptor activation e.g. from the D(E)RY motif?
- (12) Were receptor cavities filled with water before starting the simulation?
- (13) At which temperature were the simulations conducted?
- (14) Lines 231-233: The authors state the following: "This data allowed us to unambiguously assign

the identity of state P4 to a conformation of activated A_{2A}AR in the tertiary complex." If this is true, why is the state P4 also observed in DDM:CHS micelles and even predominant in POPC:POPS nanodiscs even though there is no G-protein present and therefore no ternary complex? Do you mean the state P4 is assigned to a conformation that resembles the conformation that A_{2A}AR has in the tertiary complex?

(15) Figure 5: The coloring of the bars in the histograms makes it hard to distinguish the red from the grey histogram. Maybe show the bars for each distance bin side by side instead of overlapping?

(16) Supplementary Fig. 3 is completely missing.

Reviewer #2:

Remarks to the Author:

Thakur et al. investigate the effect of membrane phospholipid composition on GPCR-G protein interactions. Building on established work on the Adenosine 2A Receptor (A_{2A}AR) in DDM:CHS micelles, the authors use ¹⁹F-NMR of a TET-labelled extrinsic cysteine residue at position 289 at the intracellular tip of transmembrane helix 7 (TM7) to characterize the conformational equilibrium of A_{2A}AR in nanodiscs. They show that in the presence of the agonist NECA and the anionic lipids POPS, POPA, POPG, and PIP2, A_{2A}AR populates a fully active conformation (P4) observed for A_{2A}AR:NECA:miniGs but not for A_{2A}AR:NECA in pure POPC nanodiscs. They show that A_{2A}AR remains fully pharmacologically active in the different nanodisc/phospholipid systems and validate lipid compositions using ³¹P NMR. Mutagenesis of selected, positively charged amino acids near the intracellular lipid-bilayer boundary to Glutamine revealed two helix 6 mutants, H230Q and K233Q that completely reversed the sensitivity of A_{2A}AR to POPS with P4 appearing with POPC but not the POPC:POPS mixture. Finally, MD simulations of inactive, fully active, and fully active A_{2A}AR with the G protein removed were conducted in POPC and POPC:POPS bilayers, supporting the authors' conclusion that G proteins couple to A_{2A}AR through an induced fit mechanism in the absence and through conformational selection in the presence of anionic lipids.

The manuscript is very well written, logically structured and contains all the necessary control experiments. The study is highly relevant and furthers the understanding of GPCR function in membrane-like environments. I highly recommend this manuscript for publication.

I have a couple of comments/suggestions:

1. Figure 1 suggests that A_{2A}AR:NECA adopts the fully active conformation (P4) in DDM:CHS micelles as well as in POPC:POPS nanodiscs. This conformation is not observed for A_{2A}AR:NECA in pure POPC containing nanodiscs. Can you comment on why you think P4 is observable in DDM:CHS but not POPC? Is A_{2A}AR more dynamic in DDM:CHS or does CHS mimic POPS?
2. Supplementary Figure 4: A) Can you comment on the horizontal movement of the CD curve from 10C to 90C and B) could it be that a potential disassembly of the nanodisc may go unnoticed because the individual MSP1D1 fragments retain alpha helicity when disassembled?
3. Page 9, starting line 185: It may be helpful to indicate the relative peak intensities in each spectrum or at least in spectra where relative peak intensities are compared.
4. Page 11, line 233: Does the smaller P4 for the miniGs complex suggest that the A_{2A}AR:miniGs interaction is transient despite the SEC suggesting a stable GPCR complex? Does A_{2A}AR in the presence of POPS have a larger P4 because POPS is present at a much higher concentration compared to the miniG and thus has a higher probability to insert into the ICL pocket?
5. Page 11, line 244: What are the POPC:anionic lipid ratios in Figure 3 (i.e. to which ratio of POPE:POPS in Supplementary Figure 10 they comparable to)? It looks like the linewidths of the POPC:POPA mixture are broader compared to other mixtures while POPC:PIP2 appears to result in generally lower intensities. Do you think this has any significance?

6. Page 13, line 271: Radioligand binding assays are only shown for K233Q and R291Q while R199Q and H230Q are missing (Supplementary Figure 11). Similarly, thermostability data is shown for the K233Q, H230Q and R291Q but not R199Q.

7. Page 13, line 288/Figure 4: P4 for H23Q and K233Q in the absence of POPS appears to be slightly upfield shifted while P4 for R291Q and possibly R199Q in the presence of POPS appears to shift slightly downfield. Do you think this has any significance?

8. I like how MD was used to rationalize the effect of POPS on the H230-R291 distance distribution. Were the distances Ca atoms measured?

The hypothesis of anionic lipids entering the G protein binding cavity is intriguing and reminded me of a study where G proteins were shown to synergistically prime a GPCR for coupling to other G protein subtypes (Gupte et al., 2017, PNAS, 114:3756).

9. Page 16, line 342: You may also want to cite Inagaki et al., 2012, JMB, 417:95. who showed that the lipid composition of nanodiscs had little effect on ligand binding NTS1 while significantly altering Gq interaction.

Minor comments:

1. Page 6, line 131: "the" appears to be missing prior to "complex with an agonist": "...antagonist and the complex with an agonist"

2. Page 8, line 178: I think this should be Figure 1.

3. Page 25, line 469: Table S1 appears to be missing.

4. Page 29, line 558: Do you mean $\sim 200\mu\text{M}$?

5. Page 32, line 633: Table SXX appears to be missing.

6. Page 33, line 652: How long were the production runs and did you run replicates? How was the MD data analyzed?

7. The EXSY spectrum shown in Supplementary Figure 7 is not discussed in the text.

Comments from the Reviewers are given below in black text, and our point-by-point response is provided below each comment in blue text. Edits are highlighted in the revised manuscript.

Reviewer #1 (Remarks to the Author):

The article "Anionic Phospholipids Control Mechanisms of GPCR-G Protein Recognition" by Thakur et al. investigates the influence of anionic lipids on various receptor states. The authors find a strong influence of membrane composition (especially anionic lipids) on the dynamic ensemble of A2AAR conformations. While the part of the manuscript describing the experimental data seems well elaborated, the MD simulations do not reach the necessary quality. In general, there is a lack of detailed presentation of the result data from the MD simulations. Some parts are even misleading. Therefore, in the current form, it is difficult to assess the validity of the simulations performed.

We appreciate the reviewer's positive assessment of the experimental work and for their recommendations. We have done significant additional work to address the reviewer's concerns, including adding requested information and carrying out additional simulations. Thanks to the reviewer's insightful critique, we feel the additional data and corrections have strengthened the paper.

Major issues:

(1) The authors should provide information on the setup (and length) of all MD-simulations. The reference in line 630 "see Table SXX for simulated systems" does lead to nowhere. What was the exact composition of the simulation (e.g. PC/PS ratio or L/P ratio)?

The missing table is now included in the Supplemental Information – please see Supplemental Table S3. We regret the oversight that left it out of the initial submission. The table includes the simulation compositions, which range from 100% POPC to 85:15 POPS:POPC. Also, the lipid-to-protein ratio has been added to the Methods, which is ~750:1.

(2) The structure of the activate receptor state (in absence of the G-protein) was restrained during the production runs. Nevertheless, distance data were extracted from these simulations even without mentioning the restraints in the main text. This is at least misleading if not a major problem.

We regret that the details of the restrained simulation were not more mentioned in the

main text. This information is now clearly stated in both the main text and the Methods section. It was certainly not our intention to mislead the reader.

We have also added new data from a set of simulations that are continuations of the restrained simulations with PS, but with the restraints turned off. These new data show the H230-R291 distribution shifts to still longer distances and looks remarkably similar to the same distribution in the fully active, mini-G α s-bound simulations. Thus, the new data provide even stronger evidence for the central hypothesis that the presence of PS headgroups pre-organizes the intracellular face for G protein coupling. We have added this data to a revised Figure 5 and expanded on this in the main text.

(3) This lack of information reduces the impact of the otherwise potentially relevant observation that PS headgroups make contact with the H230/K233/R291 cluster. Is there a way to prove this observation experimentally, e.g. by NMR?

We hope that the revisions described above and the new data alleviate the reviewer's concerns. We agree that the observation of the PS interaction with the H/K/R triad warrants closer scrutiny, and the suggestion of follow-on experiments to test it is well taken. However, this is experimentally very demanding, and the amount of work required to do this would merit an independent study. This investigation would require multiple demanding experiments and expensive samples, starting with multiple experiments with very expensive doubly (^2H , ^{15}N) and triply (^2H , ^{15}N , ^{13}C) stable-isotope labeled samples to attempt to assign one or more of those residues. Assuming one could then assign signals for these residues, which is already very challenging, one would then need to record NOESY data likely utilizing not only stable-isotope labeled receptor but also stable-isotope labeled lipids; while ^{31}P is NMR-active, it would not be sufficient on its own for carrying out such experiments. The likelihood that these experiments will yield useful information is also highly uncertain, as the ability to observe such a contact will also depend on the duration of the receptor-lipid interactions. To accomplish all this would require more than a year of work and multiple, very costly samples. This is one of the reasons we sought to explore these interactions via computational approaches that complemented our observations of the impact of lipids on the receptor conformational states.

(4) lines 355-357: The authors state the following: "In the absence of anionic lipids, A2AAR can form signaling complexes through an induced fit mechanism." A similar statement is also made in the abstract. In my opinion, this very strong statement is not fully supported by the available data. I agree that the mechanism of induced matching is relatively likely in this scenario, but why shouldn't conformational selection also be possible? It could simply be that in the absence of anionic lipids, the "correct" conformation for G protein binding is much less likely than in the presence of anionic lipids. For example, in Fig. 5b, the histograms clearly overlap, and all distances appear for both cases, just with different probabilities. Are there other data to distinguish between induced matching and conformational selection in the absence of anionic lipids?

In evaluating the role of anionic lipids, if conformational selection also contributed toward ternary complex formation, we would see evidence of this by NMR, manifested as the presence of measurable populations for peaks P4 and/or P2 for the agonist-bound receptor in absence of anionic lipids. The experimental data in Figure 2 show that for the $A_{2A}AR$ complex with agonist in the absence of anionic lipids, the population of the fully active state is vanishingly small, as evidenced by the absence of observable populations for state P4 and very little population for state P2. When agonist is bound, these peaks are clearly populated as long as there are anionic lipids present, even without the mini- G_{α_s} protein.

The experimental data showing anionic lipids induce a shift in the conformational ensemble toward active conformations is supported by the simulation data in Fig 5, which show that the H230-R291 distance is shifted toward active states by the presence of anionic lipids, and suggests that the mechanism is interaction of the anionic headgroups with specific residues.

More issues:

(1) The authors report (from line 306): ‘We noted H230, K233 and R291 form a triad that coordinates engagement with mini-Gas’. Does this interaction already exist in the underlying structure? Has it already been reported?

This interaction is present in the simulation data, signaled primarily by a distance of 4-5 Å in the H230-R291 distance in the absence of the G protein, which shifts to longer distances when engaged with either the Glu sidechain on the mini-G or an anionic lipid. The interaction of R291 with mini-G α_s is noted in the paper by Carpenter et al., Nature, 2016, reporting on the structure of the $A_{2A}AR$ complex with mini-G α_s , though this interaction is not remarked on with special significance. As far as we are aware, this particular triad has not previously been reported.

(2) From line 309: ‘The difference between inactive and fully-active G protein coupled states is signaled by the distribution of distances between H230 and R291.’ Is this just a claim of the authors? What is the rationale for this claim?

The rationale for this statement is given in the lines that follow the quoted sentence. It is based on the comparison of the H230 to R291 distance distribution in the inactive and agonist bound, mini-G bound simulations — in Fig 5a this distribution shifts to longer distances when the mini-G α_s is bound. Other pairwise distances in H/K/R do not show any such signal. In Fig 5b this shift is recapitulated without the G protein, but only in the presence of anionic headgroups.

In practice, the distance between TM6 and TM3/TM4 is more likely to be used to monitor

the activation state of the receptor, as it reflects the outward tilt of TM6.

We agree there are other structural changes that occur upon activation, including but not limited to the outward motion and rotation of TM6. Here we are focused specifically on a localized region of the receptor that is (a) what we believe is responsible for the negative charge induced shift in activity and (b) reflects the localized changes that are observed in the ^{19}F -NMR data signature of activity.

(3) In the manuscript, the authors use the term "tertiary complex", which seems rather unusual to me. I would recommend to use "ternary complex" instead.

We have replaced all instances with the term "ternary complex".

(4) Line 442: I am not sure if a distance between two residues can be 'active-like' or 'inactive-like', but it could refer to such a receptor state.

In this instance the distances observed in the simulations report on localized observables that reflect receptor states. We have clarified this in the text.

(5) Was the T4 lysozyme removed for the simulations? If so, how was the resulting gap in the structure handled?

The T4-lysozyme was removed from the inactive structure and initial coordinates for residues 209-219 were obtained from the structure of a thermostabilized mutant (PDB: 3PWH; Doré et al., Structure, 2011, 19, 1283-1293.) in which those residues are resolved. This information has been added to the Methods.

(6) How were the protein termini capped?

The termini were capped with methylamine (N-terminus) and acetyl (C-terminus) groups. This information is added to the Methods.

(7) What were the protonation states of amino acids crucial for receptor activation e.g. from the (D(E)RY motif?

The ionic lock conformation is consistent with what one expects for the active and inactive states. The distance between the closest O-N pair of Glu228 and Arg102 is about 4 Å in all of the inactive state simulations, and about 20 Å in all of the active state simulations. This is despite the fact that Glu228 is deprotonated in all cases. The uptake of a proton at Glu228 is an initial step in the activation of the receptor, which weakens the ionic lock and allows the intracellular face to open. All of our simulations of the active state are of the fully activated structure, in which H3 and H6 are far apart. Even in the unrestrained simulations of the fully active state without the G protein this distance is maintained. This

is expected since the typical trajectory length (500 nsec) is much shorter than the time needed to inactivate the receptor.

Ionic lock distances are added to Supplemental Table 3.

(8) Were receptor cavities filled with water before starting the simulation?

The receptor cavities hydrated spontaneously during the equilibration and first few tens of nanoseconds of production simulation time. The average number of internal waters ranges from 15 to 30, with the active receptor simulations tending to be more hydrated than the inactive simulations. These findings (degree of internal hydration, time required for spontaneous hydration) are broadly consistent with earlier reports, such as Jardón-Valdez, et al *Biophys J* 99:2200(2010). A snapshot of the fully active, G-protein coupled receptor is included here for the reviewer's inspection.

(9) At which temperature were the simulations conducted?

Simulations were conducted at room temperature (25 °C). This information was added to the Methods.

(10) Lines 231-233: The authors state the following: "This data allowed us to unambiguously assign the identity of state P4 to a conformation of activated A_{2A}AR in the tertiary complex." If this is be true, why is the state P4 also observed in DDM:CHS micelles and even predominant in POPC:POPS nanodiscs even though there is no G-protein present and therefore no ternary complex? Do you mean the state P4 is assigned to a conformation that resembles the conformation that A_{2A}AR has in the tertiary complex?

The observation of state P4 for the A_{2A}AR-agonist complex in DDM/CHS micelles and in the absence of the mini-G α_s protein is likely related to the fact that CHS is charged at physiological temperatures. Reviewer #2 asked a similar question and also proposed a similar rationale to their own question, which we agree with. Because this directly relates to Reviewer #2's question, we have copied our response to Reviewer #2 below:

Our interpretation of the NMR observations in DDM:CHS micelles are in line with

the reviewer's latter suggested interpretation, i.e. that the negative charge on CHS mimics the effects observed with anionic lipids. The pK_A of the carboxylic acid group of CHS is ~ 5.8 . At physiological pH values and the pH value used for our NMR study, we anticipate $>98\%$ of the CHS molecules should thus be negatively charged. Other groups have verified the charged state of CHS in earlier experiments and simulations (please see e.g., Augustyn et al., J. Phys. Chem. B, 2019, 123, 9839-9845).

Regarding the question about whether $A_{2A}AR$ is "more dynamic" in DDM:CHS, we assume the reviewer is asking about the rates of conformational exchange between the different membrane mimetics. This question is related to Supplementary Figure 7, which we had unfortunately not described in the main text in our first submission. We observe a slower rate of conformational exchange between peaks P1 and P2 in lipid nanodiscs, but do not observe differences in exchange rates for P4, either to P1 or P2, between samples prepared in DDM/CHS or lipid nanodiscs. Thus observing the manifestation of state P4 in DDM:CHS micelles is more likely in line with the presence of the negative charge on CHS.

We have added new text in the discussion section to address these important observations and provide a rationalization for them.

(11) Figure 5: The coloring of the bars in the histograms makes it hard to distinguish the red from the grey histogram. Maybe show the bars for each distance bin side by side instead of overlapping?

We have modified the histograms as suggested.

(12) Supplementary Fig. 3 is completely missing.

This was an unfortunate oversight, as the figure was present in the uploaded supporting information document but appeared not to be displayed in the combined file provided by the journal. We have corrected this issue and confirmed the presence of the figure in the revised combined file.

Reviewer 2 (Remarks to the Author):

Thakur et al. investigate the effect of membrane phospholipid composition on GPCR-G protein interactions. Building on established work on the Adenosine 2A Receptor (A_{2A}AR) in DDM:CHS micelles, the authors use ¹⁹F-NMR of a TET-labelled extrinsic cysteine residue at position 289 at the intracellular tip of transmembrane helix 7 (TM7) to characterize the conformational equilibrium of A_{2A}AR in nanodiscs. They show that in the presence of the agonist NECA and the anionic lipids POPS, POPA, POPG, and PIP2, A_{2A}AR populates a fully active conformation (P4) observed for A_{2A}AR:NECA:miniGs but not for A_{2A}AR:NECA in pure POPC nanodiscs. They show that A_{2A}AR remains fully pharmacologically active in the different nanodisc/phospholipid systems and validate lipid compositions using ³¹P NMR. Mutagenesis of selected, positively charged amino acids near the intracellular lipid-bilayer boundary to Glutamine revealed two helix 6 mutants, H230Q and K233Q that completely reversed the sensitivity of A_{2A}AR to POPS with P4 appearing with POPC but not the POPC:POPS mixture. Finally, MD simulations of inactive, fully active, and fully active A_{2A}AR with the G protein removed were conducted in POPC and POPC:POPS bilayers, supporting the authors' conclusion that G proteins couple to A_{2A}AR through an induced fit mechanism in the absence and through conformational selection in the presence of anionic lipids. The manuscript is very well written, logically structured and contains all the necessary control experiments. The study is highly relevant and furthers the understanding of GPCR function in membrane-like environments. I highly recommend this manuscript for publication. I have a couple of comments/suggestions:

We thank the reviewer for their positive assessment of our work and for their recommendations. Below, we have provided a point-by-point response to their questions and recommendations.

(1) Figure 1 suggests that A_{2A}AR:NECA adopts the fully active conformation (P4) in DDM:CHS micelles as well as in POPC:POPS nanodiscs. This conformation is not observed for A_{2A}AR:NECA in pure POPC containing nanodiscs. Can you comment on why you think P4 is observable in DDM:CHS but not POPC? Is A_{2A}AR more dynamic in DDM:CHS or does CHS mimic POPS?

Our interpretation of the NMR observations in DDM:CHS micelles are in line with the reviewer's latter suggested interpretation. The pK_A of the carboxylic acid group of CHS is ~5.8. At physiological pH values and the pH value used for our NMR study, we anticipate >98% of the CHS molecules should thus be negatively charged. Other groups have verified the charged state of CHS in earlier experiments and simulations (please see e.g., Augustyn et al., J. Phys. Chem. B, 2019, 123, 9839-9845).

Regarding the question about whether A_{2A}AR is "more dynamic" in DDM:CHS, we assume the reviewer is asking about the rates of conformational exchange between the different membrane mimetics. This question is related to Supplementary Figure 7, which we had unfortunately not described in the main text in our first submission. We observe

a slower rate of conformational exchange between peaks P1 and P2 in lipid nanodiscs, but do not observe differences in exchange rates for P4, either to P1 or P2, between samples prepared in DDM/CHS or lipid nanodiscs. Thus observing the manifestation of state P4 in DDM:CHS micelles is more likely in line with the presence of the negative charge on CHS.

We have added new text in the discussion section to address these important observations and provide a rationalization for them.

(2) Supplementary Figure 4: A) Can you comment on the horizontal movement of the CD curve from 10C to 90C and B) could it be that a potential disassembly of the nanodisc may go unnoticed because the individual MSP1D1 fragments retain alpha helicity when disassembled?

The upward trend of the curve above 65-70 °C may reflect nanodisc disassembly for a minor fraction of the sample, but the curve is not steep enough to estimate an inflection point and the majority of the nanodiscs appeared to remain assembled and soluble over the temperature range we tested. An estimated nanodisc disassembly temperature of >90 °C is consistent with earlier data reporting nanodisc disassembly temperatures ranging from 90 °C to >105 °C with binary lipid mixtures¹. We have expanded on this observation and included the reference below in our revised text.

1. Wadsäter, M., Maric, S., Simonsen, J.B., Mortensen, K. & Cardenas, M. The effect of using binary mixtures of zwitterionic and charged lipids on nanodisc formation and stability. *Soft Matter* **9**, 2329-2337 (2013).

(3) Page 9, starting line 185: It may be helpful to indicate the relative peak intensities in each spectrum or at least in spectra where relative peak intensities are compared.

We have appended Figure 1 to include bar charts comparing relative peak intensities, as we feel a quantitative comparison is appropriate to include in this figure. For Figures 2-4, we have also determined relative peak intensities and tabulated them in a new table included in the revised Supplementary Information section and expanded text related to this in the results section. Please see Supplementary Table 1 and pages 12-13.

(4) Page 11, line 233: Does the smaller P4 for the miniGs complex suggest that the A₂AAR:miniGs interaction is transient despite the SEC suggesting a stable GPCR complex? Does A₂AAR in the presence of POPS have a larger P4 because POPS is present at a much higher concentration compared to the miniG and thus has a higher probability to insert into the ICL pocket?

For the A₂AAR complex with mini-G_{α_s} in the absence of POPS, in addition to observing a diminished signal intensity for P4, we observe an increased signal intensity for state P2.

This indicates the presence of ternary complexes with a conformation that differs from the conformation corresponding to the active state P4, suggesting the absence of anionic lipids may promote an alternative conformation of the ternary complex. We have expanded on this observation in the text.

(5) Page 11, line 244: What are the POPC:anionic lipid ratios in Figure 3 (i.e. to which ratio of POPE:POPS in Supplementary Figure 10 they comparable to)? It looks like the linewidths of the POPC:POPA mixture are broader compared to other mixtures while POPC:PIP2 appears to result in generally lower intensities. Do you think this has any significance?

For most lipid mixtures in Figure 3, the nanodiscs contained molar ratios of 70:30, with POPC being the larger component. Samples with PIP2 were prepared in a ratio of 95:5 POPC:PIP2 as this is more consistent with physiologically relevant amounts of PIP2. We have added this information to a revised Figure 3 and also provided a new table tabulating the relative populations of all peaks (Supplementary Table 1).

We do observe a relatively higher integrated signal intensity for peak P4 with the sample containing PIP2 and POPC. From this observation we can conclude that PIP2 may be more effective at shifting the conformational equilibria toward an active state population than other lipids.

The line widths for the sample containing a mixture of POPC:POPA do appear to be somewhat larger. We quantitatively measured the full width at half maximum values for all spectra shown in Figure 3 and found the line widths for peaks P4 and P2 to be approximately 100-150 Hz wider than the corresponding peaks observed with other samples. These observations suggest the receptor active state exhibits a larger degree of conformational plasticity in this lipid composition. We have expanded on this observation in text added to the results section on pages 12-13.

(6) Page 13, line 271: Radioligand binding assays are only shown for K233Q and R291Q while R199Q and H230Q are missing (Supplementary Figure 11). Similarly, thermostability data is shown for the K233Q, H230Q and R291Q but not R199Q.

We have added the radioligand binding data for the R199Q and H230Q variants and have added the thermostability data for the R199Q variant. The new data further support our initial observations that all the employed A_{2A}AR variants are functional and folded.

(7) Page 13, line 288/Figure 4: P4 for H23Q and K233Q in the absence of POPS appears to be slightly upfield shifted while P4 for R291Q and possibly R199Q in the presence of POPS appears to shift slightly downfield. Do you think this has any significance?

The chemical shift difference for peak P4 between agonist bound A_{2A}AR[A289C] in

POPC/POPS lipids and H230Q in the absence of POPS is -0.09 ppm (51 Hz) and for K233Q is -0.17 ppm (97 Hz). For R199Q in the presence of POPS, peak P4 shows a chemical shift difference of +0.04 ppm (23 Hz). These differences are not large compared with the line width of P4 (~400 Hz) and the chemical shift differences between P4 and the nearest peak P3 (~2 ppm, ~1150 Hz) and thus most likely reflect subtle changes in the environment around the NMR probe independent from A_{2A}AR-lipid interactions. For this reason we focus our argument on the striking changes in the presence or absence of peak P4.

(8) I like how MD was used to rationalize the effect of POPS on the H230-R291 distance distribution. Were the distances C α atoms measured? The hypothesis of anionic lipids entering the G protein binding cavity is intriguing and reminded me of a study where G proteins were shown to synergistically prime a GPCR for coupling to other G protein subtypes (Gupte et al., 2017, PNAS, 114:3756).

The distances in Fig. 5 are between the heavy atoms at the ends of the H230 and R291 sidechains. These distances are more descriptive than the alpha carbon distances as they are the same atoms that coordinate the anionic lipid headgroup. A sentence defining the residue-residue distance has been added to the Methods.

We also thank the reviewer for pointing out the relevant reference and have included it in our revised text on page 18.

(9) Page 16, line 342: You may also want to cite Inagaki et al., 2012, JMB, 417:95. who showed that the lipid composition of nanodiscs had little effect on ligand binding NTS1 while significantly altering Gq interaction.

We thank the reviewer for pointing out this relevant paper and have included a reference to it on page 16.

Minor Comments:

(1) Page 6, line 131: “the” appears to be missing prior to “complex with an agonist”: “...antagonist and the complex with an agonist”

We have added the missing word.

(2) Page 8, line 178: I think this should be Figure 1.

We thank the reviewer for this comment and have made the correction.

(3) Page 25, line 469: Table S1 appears to be missing.

We have included the table containing the sequences of oligonucleotides used to generate the A_{2A}AR variants, which is now Table S2.

(4) Page 29, line 558: Do you mean ~200µM?

Yes, we have corrected this error.

(5) Page 32, line 633: Table SXX appears to be missing.

We have added the table (see Supplementary Table 3).

(6) Page 33, line 652: How long were the production runs and did you run replicates? How was the MD data analyzed?

The production runs/replicates are listed in the now included Supplemental Table 3. Each system was run for 500 nsec, 5 independent replicas. The only analysis is the calculation of the distance distributions as described above.

(7) The EXSY spectrum shown in Supplementary Figure 7 is not discussed in the text.

We have added new text describing the figure and discussing its relevance to the text. See pages 9 and 10.

Reviewers' Comments:

Reviewer #1:

Remarks to the Author:

The revised version of the manuscript "Anionic Phospholipids Control Mechanisms of GPCR-G Protein Recognition" by Thakur et al. is a much-improved version that essentially handles most of the raised issues. The authors now better describe the MD simulations and their setup and most details are provided.

Here are only few further comments:

- 1) For the capping of the protein, it appears that the authors confused N- and C-terminus. I guess the methylamine was at the C-terminus and acetyl at the N-terminus?
- 2) From the authors response I infer that in the active state D101 in the D(E)RY motif was not protonated. Is this correct? If yes this clearly needs to be stated in the manuscript.

Reviewer #2:

Remarks to the Author:

The authors have adequately addressed my comments and I fully support the publication of this interesting manuscript.

The only thing I noticed is that the authors didn't indicate what the numbers in parentheses in the last column (ionic lock distances) of Supplementary Table 3 refer to (SD, SEM?)